Citation: *Molecular Systems Biology* 9:655
www.molecularsystemsbiology.com

# Construction of human activity-based phosphorylation networks

Robert H Newman[1,2,21], Jianfei Hu[3,21], Hee-Sool Rho[1,4,21], Zhi Xie[3,21], Crystal Woodard[1,4], John Neiswinger[1,4], Christopher Cooper[5], Matthew Shirley[1], Hillary M Clark[1], Shaohui Hu[1,4], Woochang Hwang[3], Jun Seop Jeong[1,4], George Wu[6], Jimmy Lin[7], Xinxin Gao[1], Qiang Ni[1], Renu Goel[8], Shuli Xia[9], Hongkai Ji[6], Kevin N Dalby[10], Morris J Birnbaum[11], Philip A Cole[1], Stefan Knapp[12], Alexey G Ryazanov[13], Donald J Zack[3,5,14,15], Seth Blackshaw[4,9,16,17], Tony Pawson[18,19], Anne-Claude Gingras[18,19], Stephen Desiderio[5,17], Akhilesh Pandey[5,17], Benjamin E Turk[20], Jin Zhang[1,7,14,*], Heng Zhu[1,4,7,*] and Jiang Qian[3,7,*]

[1] Department of Pharmacology and Molecular Sciences, Johns Hopkins School of Medicine, Baltimore, MD, USA, [2] Department of Biology, North Carolina Agricultural and Technical State University, Greensboro, NC, USA, [3] Department of Ophthalmology, Johns Hopkins School of Medicine, Baltimore, MD, USA, [4] Center for High-Throughput Biology, Johns Hopkins School of Medicine, Baltimore, MD, USA, [5] Department of Molecular Biology and Genetics, Johns Hopkins School of Medicine, Baltimore, MD, USA, [6] Department of Biostatistics, Johns Hopkins Bloomberg School of Public Health, Baltimore, MD, USA, [7] The Sidney Kimmel Comprehensive Cancer Center, Johns Hopkins School of Medicine, Baltimore, MD, USA, [8] Institute of Bioinformatics, International Tech Park, Bangalore, India, [9] Hugo W. Moser Kennedy Krieger Institute, Baltimore, MD, USA, [10] Division of Medicinal Chemistry, College of Pharmacy, University of Texas at Austin, Austin, TX, USA, [11] Department of Medicine, University of Pennsylvania School of Medicine, Philadelphia, PA, USA, [12] Nuffield Department of Clinical Medicine, Structural Genomics Consortium, University of Oxford, Oxford, UK, [13] Department of Pharmacology, University of Medicine and Dentistry of New Jersey, Piscataway, NJ, USA, [14] Sol H. Snyder Department of Neuroscience, Johns Hopkins School of Medicine, Baltimore, MD, USA, [15] The McKusick-Nathans Institute of Genetic Medicine, The Johns Hopkins University School of Medicine, Baltimore, MD, USA, [16] Department of Neurology, Johns Hopkins School of Medicine, Baltimore, MD, USA, [17] Institute of Cell Engineering, Johns Hopkins School of Medicine, Baltimore, MD, USA, [18] Centre for Systems Biology, Samuel Lunenfeld Research Institute, Mount Sinai Hospital Toronto, ON, Canada, [19] Department of Molecular Genetics, University of Toronto, Toronto, ON, Canada and [20] Department of Pharmacology, Yale University School of Medicine, New Haven, CT, USA

[21]These authors contributed equally to this work.

* Corresponding authors. J Zhang or H Zhu, Department of Pharmacology and Molecular Sciences, Johns Hopkins School of Medicine, Baltimore, MD 21205, USA. Tel.: +410 502 0713; Fax: +410 955 3023; E-mail: jzhang32@jhmi.edu or Tel.: +410 502 1872; Fax: +410 502 1872; E-mail: hzhu4@jhmi.edu or J Qian, Department of Ophthalomolgy, Johns Hopkins School of Medicine, 600 North Wolfe Street, Baltimore, MD 21205, USA. Tel.: +1 443 287 3882; Fax: +1 410 502 5382; E-mail: jiang.qian@jhmi.edu

The landscape of human phosphorylation networks has not been systematically explored, representing vast, unchartered territories within cellular signaling networks. Although a large number of *in vivo* phosphorylated residues have been identified by mass spectrometry (MS)-based approaches, assigning the upstream kinases to these residues requires biochemical analysis of kinase-substrate relationships (KSRs). Here, we developed a new strategy, called CEASAR, based on functional protein microarrays and bioinformatics to experimentally identify substrates for 289 unique kinases, resulting in 3656 high-quality KSRs. We then generated consensus phosphorylation motifs for each of the kinases and integrated this information, along with information about *in vivo* phosphorylation sites determined by MS, to construct a high-resolution map of phosphorylation networks that connects 230 kinases to 2591 *in vivo* phosphorylation sites in 652 substrates. The value of this data set is demonstrated through the discovery of a new role for PKA downstream of Btk (Bruton's tyrosine kinase) during B-cell receptor signaling. Overall, these studies provide global insights into kinase-mediated signaling pathways and promise to advance our understanding of cellular signaling processes in humans.
*Molecular Systems Biology* 9: 655; published online 2 April 2013; doi:10.1038/msb.2013.12
*Subject Categories:* proteomics; signal transduction
*Keywords:* phosphorylation; signaling networks; systems biology

## Introduction

Protein phosphorylation, mediated by protein kinases, is one of the most wide-spread regulatory mechanisms in eukaryotes. Recently, several high-throughput studies designed to analyze the global properties of phosphorylation networks in various model organisms have been reported (Linding *et al*, 2007; Fiedler *et al*, 2009; Breitkreutz *et al*, 2010; Van Wageningen *et al*, 2010). Though these studies, which employed approaches based on protein–protein interactions (PPIs), genetic interactions, gene expression profiling, and motif-based predictions, have uncovered important clues about the organization and regulation of kinase-mediated signaling pathways, they are each limited in their ability to identify direct enzymatic interactions between kinases and their substrates. This requires biochemical analysis of

kinase-substrate relationships (KSRs) using purified protein components. However, global analysis of activity-based phosphorylation networks—built upon direct KSRs—is lacking in higher eukaryotes. Indeed, only ∼2000 human KSRs have been experimentally identified to date. In contrast, >70 400 *in vivo* phosphorylated serine, threonine, and tyrosine residues have been characterized by mass spectrometry (MS/MS) (Olsen *et al*, 2006; Yang *et al*, 2006; Molina *et al*, 2007; Wang *et al*, 2007; Mathivanan *et al*, 2008). Together, this implies that, for the vast majority of identified *in vivo* phosphorylation sites, the specific kinase(s) responsible for the phosphorylation event remains unknown.

## Results

To help narrow this knowledge gap, we developed a new strategy based on functional protein microarrays and bioinformatics analysis to assign upstream kinases to specific phosphorylation events found *in vivo*. This strategy, which we have dubbed 'CEASAR' because it provides a general framework for *C*onnecting *E*nzymes *A*nd *S*ubstrates at *A*mino acid *R*esolution, was used to construct a high-resolution map of human phosphorylation networks that connects kinases to specific phosphorylation sites on their downstream substrates. In addition to *in vivo* phosphosites, such a map requires two key elements: (1) an activity-based phosphorylation network based on direct KSRs and (2) information about the consensus phosphorylation motif of each kinase in the network. To this end, we first employed human protein microarrays to experimentally determine substrates for 289 unique human kinases (Supplementary Table 1). We then developed a new algorithm, based on both the experimentally derived KSRs and *in vivo* phosphorylation sites identified by MS/MS, to determine phosphorylation motifs for each kinase in the collection. Finally, we combined these KSRs, *in vivo* phosphosites, and the newly determined motifs to connect kinases to specific phosphosites, resulting in a high-resolution map of human phosphorylation networks (Figure 1). Application of this map led to the discovery of a new role for cAMP-dependent protein kinase (PKA) downstream of Bruton's tyrosine kinase (Btk) during B-cell receptor (BCR) signaling. We envision that the CEASAR strategy can be applied to additional data sets to generate high-resolution maps of other protein post-translational modifications important to cellular physiology.

### Identification of human KSRs using protein microarrays

The protein microarrays used during this study are composed of 4191 unique, full-length human proteins representing 12 major protein families (Hu *et al*, 2009). Among the protein families that are represented on the microarrays, some, such as transcription factors (TFs), kinases and RNA-binding proteins, are known to be widely regulated by phosphorylation, while in others, such as mitochondrial proteins, the role of protein phosphorylation is less well understood (Supplementary Figure 1). To identify those proteins on the array that could be phosphorylated by a given kinase, individual protein

microarrays were incubated with active, full-length human kinases in the presence of [γ-$^{32}$P]ATP, as described previously (Zhu *et al*, 2000; Ptacek *et al*, 2005; Supplementary Figures 2 and 3; see Materials and methods). For each batch of phosphorylation reactions, one microarray was also incubated in the absence of any kinase to identify those proteins that underwent autophosphorylation and/or bound ATP tightly (Supplementary Figure 4). These control experiments led to the removal of 52 proteins from further analyses due to autophosphorylation or direct binding to ATP. The reproducibility of the phosphorylation assay was confirmed by performing a subset of the phosphorylation reactions in duplicate (Supplementary Figure 5). To identify the specific substrates of a given kinase, the proteins on the array were scored using an algorithm designed to measure the relative signal intensity of each spot (Hu *et al*, 2009). Using a cutoff value of three standard deviations above the mean intensity (Supplementary Figure 6), we identified 24 046 phosphorylation events involving 289 unique kinases and 1967 unique substrates. This collection of *in vitro* 'hits' was termed as the 'rawKSR' data set (Figure 1; Supplementary Table 2).

To enrich the KSRs for physiologically relevant processes, we applied a Bayesian statistics model to the rawKSR data set (see Materials and methods). For this analysis, we hypothesized that a kinase and its physiologically relevant substrate(s) are more likely to share similar tissue expression patterns, localize to the same subcellular compartment, and/or physically interact with one another, either directly or indirectly (Supplementary Figure 7). We then assembled two training sets: (1) a positive set composed of 1103 known KSRs curated from the literature and (2) a negative set devoid of known protein kinases. Comparing the positive and negative training sets, we determined relative weights for the above three features and calculated the likelihood (*L* score) for each of the 24 046 KSRs. Using a *P*-value of 0.05 as a threshold, we predicted 3656 refined KSRs (refKSRs)—involving 255 unique kinases and 742 substrate proteins—that were most likely to be physiologically relevant (Supplementary Table 3).

### Evaluation of refKSRs

Three lines of computational evidence suggest that the Bayesian analysis improves the fidelity of our refKSR data set (Supplementary Figure 8). First, the percentage of phosphoproteins that have been confirmed *in vivo*—based mainly on global MS/MS analysis—was significantly improved in the refKSR list, increasing from 66% (1291/1967) in the rawKSR data set to 77% (567/741) in the refKSR data set ($P = 7.87 \times 10^{-16}$). Second, cross-validation analysis revealed that the Bayesian approach increases the recovery rate of known KSRs over five-fold ($P = 1.37 \times 10^{-15}$). Third, the enriched functions of the identified substrates for individual kinases showed better agreement with the known functions of their upstream kinases. Specifically, known functions were recovered for 53 kinases based on the enriched functions of their respective substrates, a 2.4-fold improvement over the rawKSR data set ($P = 2.1 \times 10^{-11}$). Taken together, these findings suggest that the refKSR set, as compared with the rawKSR set, is significantly improved with regard to its physiological relevance.

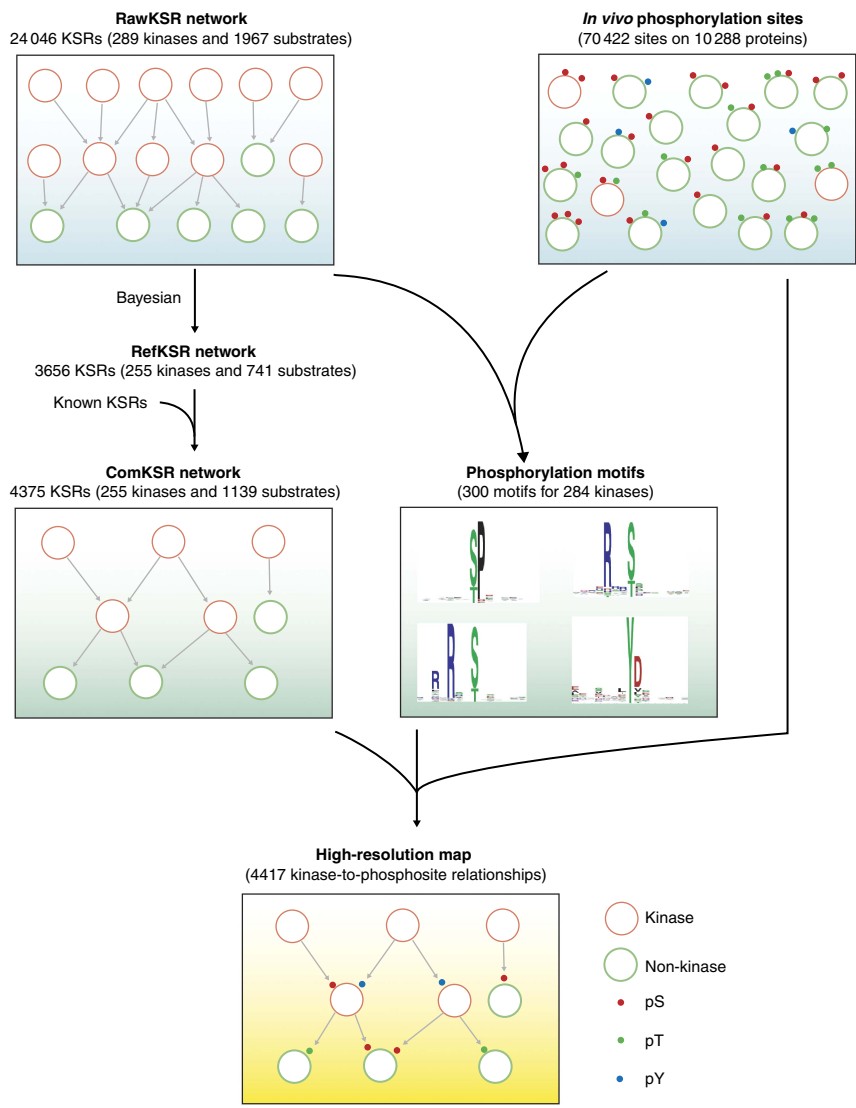

**Figure 1** Schematic diagram of the CEASAR strategy. The rawKSR data set (upper, left panel) is composed of 24 046 KSRs identified *in vitro* using purified human kinases and functional protein microarrays. This data set was used as a starting point to create a high-resolution map of human phosphorylation networks using the CEASAR strategy. First, to identify those KSRs that are likely to occur under physiological conditions, Bayesian network analysis of known KSRs was used to derive an algorithm that assigned a likelihood score to each of the experimentally derived KSRs in the rawKSR data set. This information was then used to construct a refined KSR (refKSR) data set composed of 3656 novel KSRs likely to occur under physiological conditions. Finally, the refKSRs were combined with 719 known KSRs to generate the combined KSR (comKSR) data set. The comKSR data set (middle, left panel), which consists of 4375 KSRs, was used to construct the human phosphorylation network upon which the high-resolution map is built. Next, the rawKSR data set was combined with information about *in vivo* sites of phosphorylation (upper, right panel) to determine consensus phosphorylation motifs using the M3 algorithm. Using this approach, we identified consensus motifs for 284 of the 289 kinases in our collection (middle, right panel). Finally, information about consensus sites and *in vivo* sites of phosphorylation were integrated with the comKSR data set to yield a high-resolution map of human phosphorylation networks (bottom panel). This network, which connects 4417 phosphosites on substrates to their cognate kinase, includes only those sites that could be unambiguously assigned to a given kinase. Phosphoserine (pS), phosphothreonine (pT), and phosphotyrosine (pY) sites are denoted as red dots, green dots, and blue dots, respectively.

To experimentally evaluate the refKSRs, we randomly selected 243 KSRs, involving 75 kinases and 136 substrates, and tested their relationships in transfected cells (Supplementary Table 4; Supplementary information). To this end, a vector encoding a FLAG-tagged version of each substrate was transfected into HeLa cells in the presence of either a V5-tagged version of the cognate kinase or an empty vector. The substrates were then assayed for kinase-dependent changes in their electrophoretic mobility and/or protein levels inside cells (Figure 2A). Of the 243 KSRs tested, 71% showed

detectable substrate expression, among which kinase-dependent changes were observed in 76% of the cases (Supplementary Figure 9). In contrast, for a negative control set where no obvious phosphorylation signals were detected in the microarray assays involving the kinase-of-interest, we observed changes in <10% of the pairs (4/42).

Among the substrates that exhibited kinase-dependent changes in the co-transfection assay, the most common change observed was altered protein levels (39 and 46% of the substrates exhibited either an increase or a decrease in protein

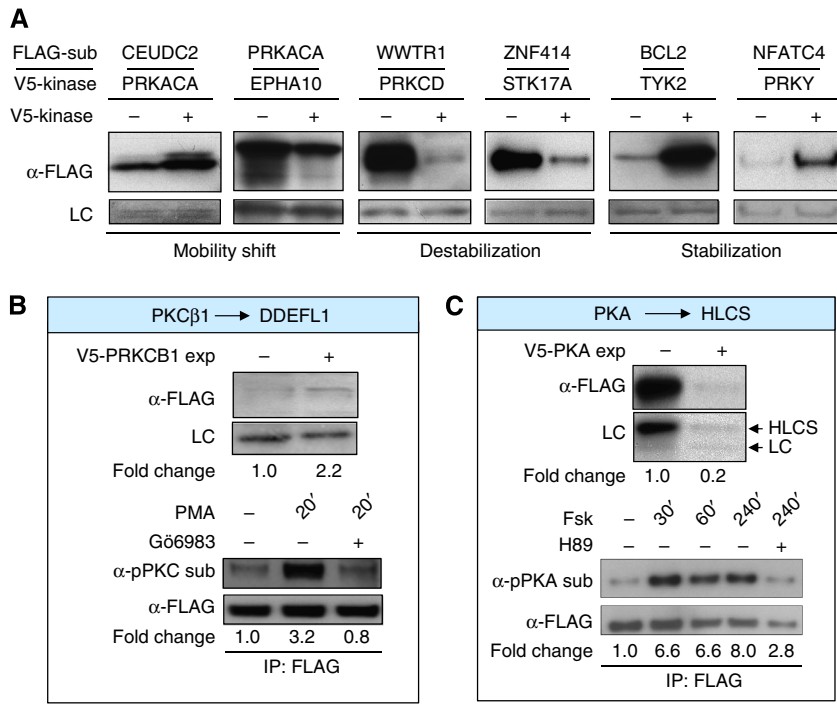

**Figure 2** Cell-based validation of refKSRs. (**A**) Examples from the first round of cell-based validation experiments. FLAG-tagged substrates were co-transfected with either a V5-tagged version of the cognate kinase or an empty vector and assayed for kinase-dependent changes in electrophoretic mobility and/or protein levels. Three major kinase-dependent changes were observed in the substrates: mobility shift, substrate depletion (destabilization), or substrate accumulation (stabilization). LC, loading control. (**B**, **C**) Two examples from the second round of cell-based validation studies. (B) Top panel: PKC-dependent stabilization of DDEFL1/ASAP3. FLAG-DDEFL1 was expressed in the presence or absence of V5-PRKCB1, as described in (A). Bottom panel: HeLa cells transfected with FLAG-DDEFL1 were treated with phorbol-12-myristate-13-acetate (PMA) for the indicated times in the presence or absence of the PKC inhibitor, Gö6983. FLAG-DDEFL1 was then immunoprecipitated and probed with an antibody specific for phosphorylated PKC substrates before being stripped and re-probed with an anti-FLAG antibody. The normalized intensity ratio for each band is shown below the lane. (**C**) Top panel: PKA-dependent stabilization of HLCS. FLAG-HLCS was expressed in the presence or absence of V5-PRKACA, as described in (A). Bottom panel: HeLa cells transfected with FLAG-HLCS were treated with forskolin (Fsk) in the presence or absence of the PKA inhibitor, H89, for the indicated times. FLAG-HLCS was then immunoprecipitated and probed with an antibody specific for phosphorylated PKA substrates before being stripped and probed with an anti-FLAG antibody. The normalized intensity ratio of each band is shown below the corresponding lane. Overall, 15 of the 19 refKSRs tested were validated.

levels in the presence of kinase, respectively). These data suggest that many kinases may control protein stability, either directly or indirectly. To determine whether the observed changes in protein stability were due to direct phosphorylation of the substrate by its cognate kinase, a second round of validation experiments were conducted. During these experiments, we chose to focus on the action of endogenous kinases. Therefore, we selected 19 KSRs in which the kinases were restricted to protein kinase C (PKC), PKA, protein kinase B (PKB/Akt), and extracellular signal-regulated kinase 1 (Erk1/MAPK3). These kinases were chosen because (1) their endogenous activity can be both induced and inhibited pharmacologically and (2) the phosphorylation of their substrates can be readily detected using commercially available antibodies. For instance, in the first round of validation experiments, we observed that DDEFL1/ASAP3, a GTPase activating protein involved in cell differentiation and migration (Ha *et al*, 2008), underwent a PKC-dependent increase in its protein levels (Figure 2B, top panel). To determine whether these changes are correlated with phosphorylation by PKC, we monitored the extent of PKC-mediated phosphorylation on FLAG-tagged DDEFL1/ASAP3 in HeLa cells in the presence of the PKC activator,

phorbol-12-myristate-13-acetate (PMA). Within 20 min of PMA treatment, PKC-mediated phosphorylation on immuno-precipitated DDEFL1/ASAP3 increased >3-fold, as detected by an antibody that specifically recognizes phosphorylated PKC substrates. Importantly, this phenomenon could be completely inhibited by pre-incubation with the PKC inhibitor, Gö6983, suggesting that endogenous PKC directly phosphorylates DDEFL1/ASAP3 in cells. Interestingly, recent evidence suggests that DDEFL1/ASAP3, which is known to regulate the GTPase, ADP-ribosylation factor (Arf), may be involved in cross-talk between the Arf and $Ca^{2+}$ signaling pathways (Ismail *et al*, 2010).

In another case, co-expression of PKA caused a reduction in the levels of holocarboxylase synthetase (HLCS), an essential biotin ligase involved in chromatin remodeling and several metabolic processes (Figure 2C, top). Closer examination of this interaction revealed that PKA-mediated phosphorylation of HLCS occurred <30 min after the addition of forskolin, a pharmacological activator of the cAMP/PKA pathway (Figure 2C, bottom). This phenomenon was largely attenuated by pre-treatment of the cells with the PKA inhibitor H89 (Figure 2C, bottom), supporting the notion that HLCS is specifically phosphorylated by PKA.

Overall, 15 of the 19 KSRs tested (79%) were confirmed in the second round validation, suggesting that in most cases the observed differences in substrate protein levels reflected direct phosphorylation by their corresponding kinase (Supplementary Table 5; Supplementary Information). Taken together, our computational and experimental evaluations indicate that the refKSR data set is of high quality. To take advantage of the existing knowledge base, 741 known KSRs curated from the literature were integrated with the 3656 refKSRs described above to generate the combined (comKSR) data set (Supplementary Table 6). This data set provides a foundation for gaining new insights into the organization and function of human phosphorylation networks.

## Identification of phosphorylation motifs

As the next step of the CEASAR strategy, we developed an integrated algorithm, termed M3 (*M*otif discovery based on *M*icroarray and *M*S/MS), to systematically identify phosphorylation motifs (Figure 3A; see Materials and methods). This approach combines our KSR data with *in vivo* phosphorylation sites determined primarily by MS/MS analysis and extracts motifs based on an iterative procedure. Because kinase-substrate recognition is a biochemical property, we predicted the phosphorylation motifs based on the 24 046 rawKSRs as well as the 719 known KSRs from the comKSR data set. To this end, 13 244 of the 70 422 phosphorylation sites identified by MS/MS were mapped to 1644 substrates found in the rawKSR and comKSR data sets. Short amino-acid sequences (i.e., 15-mers) centered about these phosphosites were then binned into groups according to the identified KSRs. Though it is possible that some of the sequences within a given group contained phosphorylation sites that are recognized by several kinases, we assumed that, among all of the 15-mers for a particular kinase, those sequences recognized by the kinase-of-interest carry a statistically enriched consensus motif. Therefore, M3 is designed to utilize an iterative approach to identify statistically enriched consensus motifs within each group (Figure 3A; Supplementary Methods).

For a given kinase, the iterative method begins with a matrix representing the relative occurrence of each amino acid at a particular position in the matrix from among a group of identified substrates (containing $n$ phosphopeptide sequences) (Step 1; Figure 3A). Each phosphopeptide sequence is given a score based on how well it matches to the initial matrix. The top 10 sequences are then grouped as seed sequences and a position weight matrix (PWM) is generated (Step 2). The remaining sequences ($n - 10$) are then compared with this PWM to identify the top-matched sequence. This sequence is then added to the seed sequences and the PWM is updated (Step 3). The entire process is repeated until the best score of the remaining sequences is below a cutoff, which is determined based on the distribution of matching scores for random sequences, or until the number of seed sequences is equal to the number of substrates of the kinase determined during the phosphorylation assay (Step 4). In the case of dual-specificity kinases, we separately considered motifs that contained pS/T or pY sites. Using this approach, we identified 300 consensus motifs for 284 human kinases, representing 55% of the human kinome (see Supplementary Information for the PWM of the 300 motifs).

To independently validate the identified phosphorylation motifs, we compared our predicted phosphorylation motifs for 24 kinases with those obtained using a positional scanning peptide library (Hutti *et al*, 2004; Mok *et al*, 2010). Comparison of PWMs of the motifs identified by the two approaches revealed that 75% (18/24) of the motifs were significantly similar to one another (Figure 3B; Supplementary Figure 10). This high correlation stands in contrast to a randomized motif set, which yielded only 5% matching motifs above the same cutoff (Supplementary Figure 11). Furthermore, a comparison with the literature recovered 48 additional motifs that resemble those predicted using a different approach (Supplementary Figure 12; Miller *et al*, 2008). These results suggest that the motifs identified by the M3 approach are reliable.

## A high-resolution map of human phosphorylation networks

Finally, to create a high-resolution map of human phosphorylation networks that joins each kinase in the network to its downstream substrates at specific phosphorylated residues, we integrated the information about both phosphorylation motifs and *in vivo* phosphorylation sites into our comKSR data set. The resulting phosphorylation map connects 230 kinases to 2591 *in vivo* phosphorylation sites in 652 substrates, representing 4417 kinase-to-phosphorylation site relationships (see Supplementary Information for a Cytoscape session file illustrating the phosphorylation networks). While 758 phosphorylation sites with known upstream kinases were correctly connected to their respective kinases, the other 3659 relationships represent newly identified connections (Figure 4A).

To experimentally evaluate the fidelity of these newly identified connections, we examined what effect mutation of the predicted phospho-acceptor site had on the substrates in the presence of kinase for three selected KSRs (see Supplementary Information). For example, using the cell-based assay described above, we observed that the PKA-dependent increase in DAXX protein levels was largely abolished when Ser688 was mutated to Ala (DAXX[S688A]) (Figure 4B). Similar phenotypes were also observed for the other two sets (Figure 4B). To evaluate the sensitivity of our predictions, we also selected a negative control set consisting of the KSR, PAK1→DAXX. Though it recognizes a consensus motif (RxS) that is similar to that of PKA (RxxS) and also promotes the accumulation of DAXX, PAK1 is not predicted to phosphorylate DAXX on Ser688 in the high-resolution map. Consistent with this notion, the S688A mutation had no effect on DAXX protein levels in the presence of PAK1 (Figure 4B), suggesting that PAK1 promotes the accumulation of DAXX by phosphorylation of a residue(s) other than S688. To determine whether mutation of the predicted residues has a direct impact on the extent of kinase-mediated phosphorylation, we conducted *in vitro* phosphorylation reactions using purified wild-type (WT) and mutated substrate proteins. As illustrated in Figure 4C, the two predicted site-specific KSRs, as well as the

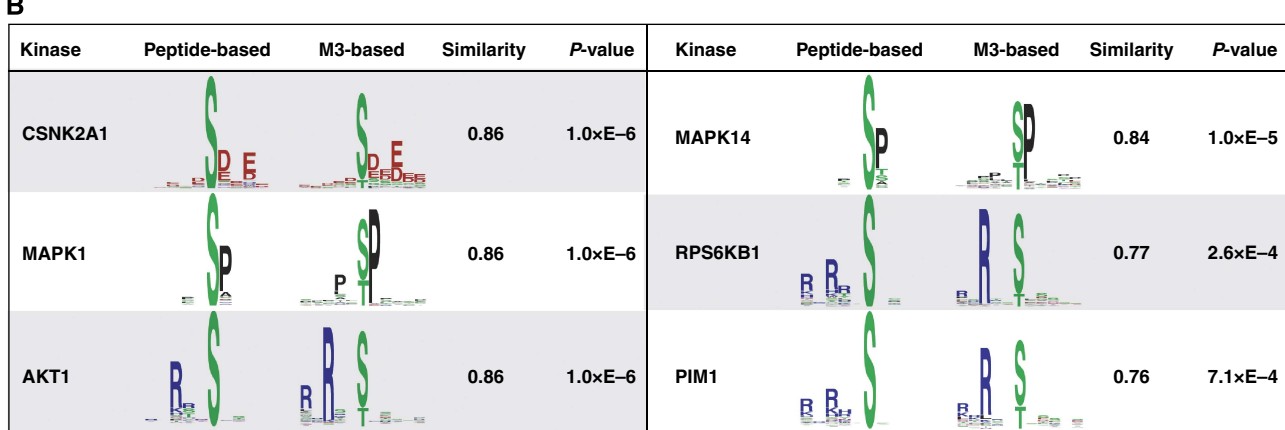

**Figure 3** Identification of phosphorylation motifs by M3. (**A**) To predict consensus phosphorylation motifs for the kinases in our collection, we first obtained all known phosphorylation sites on the substrates of a given kinase (e.g., CAMK2D) as foreground. Likewise, the phosphorylation sites in all human proteins were collected as background (Step 1). Each site in the foreground was then compared with the observed and expected frequency matrix and the sites exhibiting the best scores were included as seed sites (Step 2). The foreground frequency matrix was then updated with the seed sequences and the remaining sequences compared with this matrix to identify the top-matched sequence. The resulting sequence was then added to the seed sequences and the matrix was updated (Step 3). The entire process was repeated until the best score of the remaining sequences was below a cutoff or until the number of seed sequences was equal to the number of substrates of the kinase determined during the phosphorylation assay (Step 4 and Step 5). The final set of sites was used to derive the consensus sequence of the kinase (Step 6). (**B**) Comparison between motifs generated using M3 and scanning peptide array approaches. Representative motifs identified by M3 (right) and peptide library (left) approaches are shown. For each example, the similarity score between the motifs generated using the two methods, along with the corresponding *P*-values, are tabulated to the right.

PAK1→DAXX-negative control, behaved as predicted. Taken together, these results suggest that our high-resolution map of human phosphorylation networks is of high quality.

To our knowledge, this is the first map of human phosphorylation networks at amino-acid resolution based solely on experimental data.

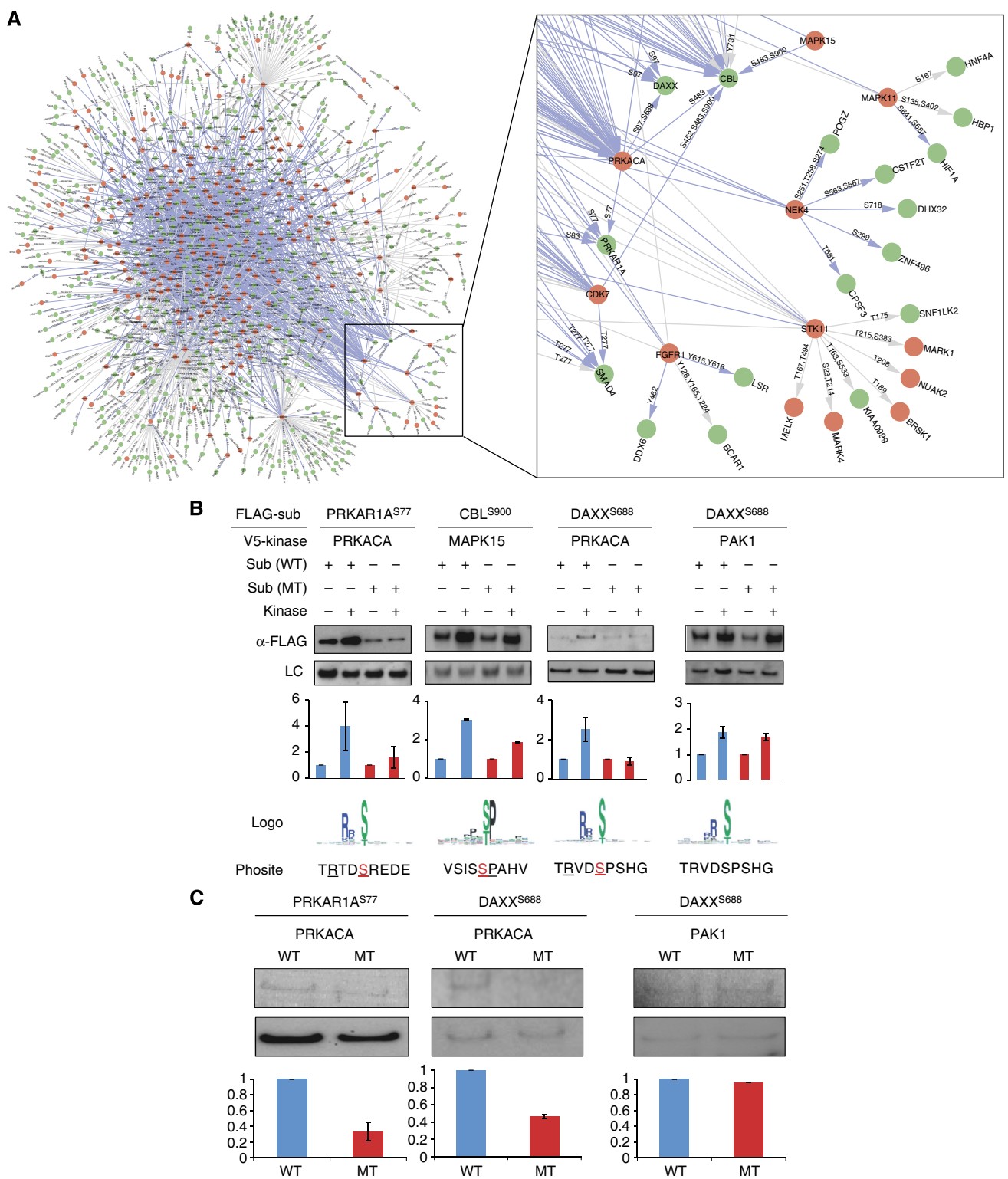

**Figure 4** A high-resolution map of the human phosphorylation network. (**A**) A high-resolution map in which *in vivo* phosphorylation sites, predicted to be phosphorylated by an upstream kinase(s), are annotated on the edges representing the KSRs (see inset). Blue edges represent the KSRs obtained in this study, while gray edges denote KSRs curated from the literature. Orange and green nodes represent kinase and non-kinase proteins, respectively. A searchable version of this network can be found at phosphonetworks.org. (**B**) Cell-based validation of phosphorylation site predictions. The mutated site is indicated by superscript. For each lane, the signal intensity (α-FLAG) was first normalized to that of the loading control (LC) and then compared with the 'substrate alone' signal for each set. The blue and red bars represent WT and mutant (MT) sets, respectively. Error bars represent standard error of at least two independent experiments. The predicted phosphorylation motif (Logo) and the primary amino-acid sequence surrounding the mutated site within each substrate (Phosite) are indicated. Those residues that are predicted to contribute to recognition by the kinases are underlined and the mutated site is highlighted in red. (**C**) *In vitro* validation of phosphorylation site predictions. The signal intensity of each radio-labeled band is plotted as in (B). Error bars represent the standard error of at least two independent experiments.

## Identification of intermediate kinases in signaling pathways

Despite recent advances in the global analysis of cellular signaling pathways, our understanding of most signaling networks is still largely incomplete due to the existence of many as yet unidentified components and indirect connections within these networks. We hypothesized that, based on the large number of new KSRs identified in this study, we could identify such 'missing links' by superimposing our data onto annotated signaling pathways derived from the literature or databases (e.g., KEGG). Moreover, when combined with information derived from our high-resolution phosphorylation map, this type of analysis can offer insights into the physiological consequences of the specific phosphorylation events underlying these connections. For instance, Btk is known to play an important role during BCR signaling while its binding partner, ARID3A, has been implicated in the transcriptional regulation of the *IgH* locus (Lin *et al*, 2007). Recent studies suggest that functional Btk is required for ARID3A activity (Rajaiya *et al*, 2005). However, no tyrosine phosphorylation sites have yet been identified in ARID3A, suggesting that Btk likely does not phosphorylate ARID3A directly (Rajaiya *et al*, 2006). Based on our refKSR data set, we identified PKA as both a substrate of Btk and an upstream kinase of ARID3A, raising the possibility that PKA serves as an intermediary between Btk and ARID3A.

To test this hypothesis, we first validated the Btk→PKA and PKA→ARID3A KSRs in HeLa cells using the cell-based assays described above. In the case of PKA→ARID3A, either ectopic expression of PKA or activation of endogenous PKA led to increases in both ARID3A protein levels and the extent of PKA-mediated phosphorylation on ARID3A (Figure 5A–C; Supplementary Figure 13A; Supplementary Information). Based on the high-resolution phosphorylation map, we identified two putative sites of PKA-mediated phosphorylation on ARID3A: a strong consensus site at S353 and a weaker one at S333 (Supplementary Figure 13B). Interestingly, though both sites appear to be phosphorylated by PKA inside cells, only S353 contributes to the stabilization phenotype (Figure 5D; Supplementary Figure 13B).

On the other hand, co-expression of Btk with PKA resulted in a change in PKA's migration pattern suggestive of PKA activation (Figure 5E). Consistently, overexpression of Btk alone caused a global increase in phosphorylated PKA substrates (Supplementary Figure 13C) and phenocopied PKA with respect to its effect on ARID3A protein levels (Figure 5B), suggesting that Btk-mediated phosphorylation might enhance PKA's kinase activity. To determine how Btk-mediated Tyr phosphorylation might affect the activity of PKA, we first demonstrated that Btk preferentially phosphorylates PKA on Y331 *in vitro*, as predicted by the high-resolution map (Figure 5F). We then showed that mutation of this site to Phe (PKA$^{Y331F}$) completely abolishes the ability of Btk to enhance PKA-mediated phosphorylation of ARID3A (Figure 5G). Together, these data suggest that Btk directly phosphorylates PKA on Y331, leading to enhanced PKA kinase activity.

To further characterize this connection in a more physiologically relevant context, we examined the relationships between endogenous Btk, PKA, and ARID3A during BCR

signaling in Ramos B cells. First, we observed that Tyr phosphorylation of PKA, which increased ~2-fold 10 min after BCR activation, was prevented by pre-treatment of the B cells with the Btk-selective inhibitor, terric acid (TA) (Kawakami *et al*, 1999; Figure 5H). The observed increase in Tyr phosphorylation on PKA correlated with an increase in the extent of PKA-mediated phosphorylation on a portion of cellular PKA substrates, including ARID3A (Figure 5I and J). Importantly, ARID3A phosphorylation was inhibited by pre-treatment with either H89 or TA, suggesting that this phenomenon is both PKA and Btk dependent (Figure 5J). Moreover, PKA-mediated phosphorylation appears to promote the accumulation of ARID3A in B cells, as evidenced by a substantial increase in ARID3A levels <1 h after BCR activation (Supplementary Figure 13D).

Interestingly, aside from its effect on ARID3A protein levels, we found that PKA activation by Btk may also have other important roles during BCR signaling. For instance, pre-treatment of B cells with H89 both slowed the onset and reduced the magnitude of $Ca^{2+}$ release from intracellular stores in a dose-dependent manner following BCR activation, leading to a pronounced reduction in the extent of $Ca^{2+}$ influx (Figure 5K). This is consistent both with Btk's known role in the regulation of $Ca^{2+}$ signaling downstream of the BCR and with PKA's ability to regulate $Ca^{2+}$ dynamics in other cell types (Ni *et al*, 2011).

Taken together, our in-depth characterization of PKA as the missing link between Btk and ARID3A demonstrated a new mode of enhancing PKA activity via Tyr phosphorylation by Btk, as well as a potentially new role for PKA during BCR signaling.

## Discussion

Although other high-throughput approaches, such as yeast two-hybrid, TAP tag-coupled MS/MS, and synthetic genetic screening, have been used to construct phosphorylation networks, the edges in the networks generated by these methods do not necessarily represent direct KSRs (Zheng *et al*, 2000; Olsen *et al*, 2006; Yang *et al*, 2006; Molina *et al*, 2007; Wang *et al*, 2007; Mathivanan *et al*, 2008; Fiedler *et al*, 2009). Furthermore, though various shotgun MS/MS approaches have identified an extremely large number of phosphorylated residues in mammals, the immediate upstream kinases targeting these sites have not been experimentally determined in most cases. To address these challenges, we developed an effective strategy, termed CEASAR, which combines experimentally derived KSRs obtained from phosphorylation reactions performed on protein microarrays with sophisticated data integration and thorough validation to generate a high-resolution map of human phosphorylation networks. Importantly, our activity-based networks complement and even extend the information content provided by many of the approaches alluded to above. This synergy is demonstrated both by the development of the M3 algorithm, which combines microarray data with MS data to predict consensus phosphorylation motifs, and by the construction and application of the high-resolution phosphorylation map, which combines the information about KSRs with phosphorylation motifs and

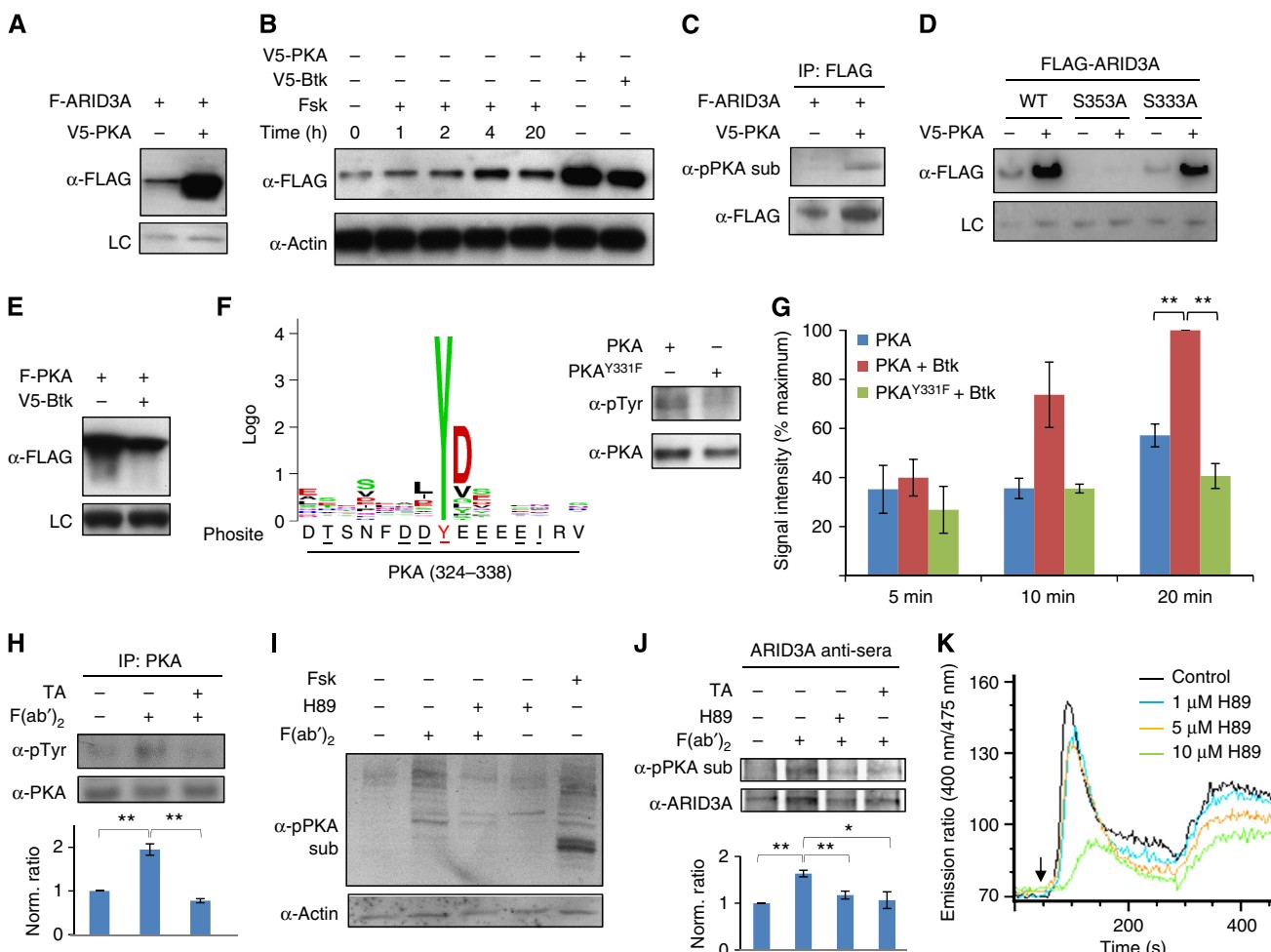

**Figure 5** Characterization of PKA as a missing link between Btk and ARID3A. (**A**) Kinase-dependent changes in FLAG-tagged ARID3A (F-ARID3A) protein levels in the presence or absence of V5-PKA in HeLa cells. LC, loading control. (**B**) Effect of endogenous PKA activation on ARID3A protein levels. (**C**) Validation of the PKA-ARID3A KSR using coupled IP-immunoblot analysis in HeLa cells. α-pPKA sub, an antibody against phosphorylated PKA substrates. (**D**) Identification of S353 as the phosphorylation site that contributes to PKA-mediated stabilization of ARID3A. (**E**) Validation of the BTK-PKA KSR in HeLa cells. (**F**) Identification of PKA Y331 as a Btk phosphorylation site. Left: Btk phosphorylation motif (Logo) generated by the M3 algorithm and the predicted site of Btk-mediated phosphorylation on PKA. Formatting is as described in Figure 4B. Right: *In vitro* kinase assays using recombinant Btk and WT PKA and PKA[Y331F] (MT) as substrates. The extent of Tyr phosphorylation and total PKA in each lane was determined using antibodies against pTyr residues (α-pTyr) and PKA (α-PKA), respectively. (**G**) Effect of Btk-mediated phosphorylation on PKA activity *in vitro*. Recombinant WT PKA was first incubated in the presence (red) or absence (blue) of Btk in reaction buffer containing cold ATP. After Btk removal, PKA was then incubated with recombinant ARID3A in the presence of [γ-$^{32}$P]-ATP and phosphorylated ARID3A was measured. The same assay was repeated using mutant PKA (PKA[Y331F]) in the presence of Btk (green). $n = 2$–4 trials per condition $\pm$ s.e.m.; two-tailed *t*-test versus PKA + Btk, PKA: **$P = 5.7 \times 10^{-5}$, MT + Btk: **$P = 2.2 \times 10^{-5}$. (**H**) Btk-mediated phosphorylation of endogenous PKA following BCR activation. Each value represents the average pTyr signal intensity, normalized against total PKA. TA, terric acid. $n = 3$ per condition $\pm$ s.e.m.; one-tailed *t*-test versus F(ab′)$_2$ alone ($-/+$), untreated ($-/-$): **$P = 0.009$, F(ab′)$_2$ + TA ($+/+$): **$P = 0.004$. (**I**) Effect of BCR activation on PKA activity. (**J**) PKA-mediated phosphorylation of ARID3A following BCR activation. The average pPKA substrate signal intensity, normalized against total ARID3A, is shown below each lane. $n = 4$ per condition $\pm$ s.e.m.; one-tailed *t*-test versus F(ab′)$_2$ alone ($-/-/+$), untreated ($-/-/-$): **$P = 0.003$, F(ab′)$_2$ + H89 ($-/+/+$): **$P = 0.007$, F(ab′)$_2$ + TA ($+/-/+$): *$P = 0.019$. (**K**) Effect of PKA inhibition on Ca$^{2+}$ dynamics following BCR activation. Representative calcium traces of F(ab′)$_2$-stimulated Ramos B cells in the presence of various concentrations of H89 were obtained using Indo-1 imaging. The addition of F(ab′)$_2$ is indicated by an arrow.

*in vivo* sites of phosphorylation to predict specific sites of phosphorylation on a given kinase substrate.

Several lines of evidence suggest that our high-resolution map of phosphorylation networks is of high quality. For instance, because 758 of 1156 known site-specific kinase-substrate interactions were recovered by our method, the false negative rate is $1 - (758/1156) = 34.4\%$ and the recovery rate is $758/1156 = 65.6\%$. That being said, the possibility exists that the sites predicted to be phosphorylated on a given substrate are not necessarily the sites targeted by the kinase-

of-interest during the protein microarray experiments. For instance, the true target site(s) may not yet be identified by MS/MS. Nevertheless, based on our validation experiments, each of the 5 kinase-phosphosite interactions examined were confirmed by protein mutagenesis experiments (including those phosphosites identified in Figures 4 and 5), suggesting that the false positive rate is rather low. The quality of our networks is further demonstrated when we compared them with other existing phosphorylation networks. For example, NetworKIN, a phosphorylation network based on motifs

predicted by scanning peptide arrays, included 7143 site-specific kinase-substrate interactions (Linding *et al*, 2007). After removing those kinase-substrate pairs for which either the kinase or the substrate was associated with an outdated ENSEMBL ID, 6338 site-specific kinase-substrate interactions remained. Among these, 48 are known interactions, suggesting that the true positive rate for NetworKIN is 0.76% (48/6338). In contrast, our data recovered 758 known interactions, with a true positive rate of 17.2% (758/4417). We believe that the >20-fold improvement in the true positive rate of our networks can be, at least partially, explained by the fact that we used full-length proteins, rather than peptides, both to build the network and to extract motifs. We consider this to be a major advantage of the CEASAR strategy. For instance, kinase activities are constrained by substrate accessibility within the fully folded protein structure. Moreover, additional protein interactions, which are absent in short peptide sequences, often have important roles in substrate recognition. On the other hand, there are also potential deficiencies of using purified proteins, including improper folding, artificial steric hindrance or the absence of auxiliary factors, such as scaffolding proteins. Together, these factors might contribute to the high false negative rate observed in the refKSR data set (~95%).

It is interesting to note that, among those substrates that exhibited a kinase-dependent change during the first round of validation experiments, the majority of them (112/132 = ~85%) were associated with changes in protein stability. We believe that one explanation for this observation may lie, in part, in the fact that a large proportion of the substrates in our refKSR data set are composed of TFs and their co-regulators. Indeed, phosphorylation-dependent degradation/stabilization of TFs is well documented and may be a common mode of regulation for this class of proteins (Pahl and Baeuerle, 1996; Whitmarsh and Davis, 2000; Gao and Karin, 2005). In support of this notion, over half (~55%) of the substrates that underwent kinase-dependent changes in stability were TFs, despite the fact that only ~35% of the test set were from this family. As the knowledge base grows, it may be possible to unambiguously identify sites of phosphorylation on many of the substrates in our refKSR data set (thereby expanding the number of kinase-phosphosite relationships present in the high-resolution map). In those cases where the substrate undergoes kinase-dependent changes in stability, it will be interesting to see if sequence motifs involved in the regulation of protein stability, such as phospho-degrons (in the case of degradation) (Dinkel *et al*, 2012) or a SUMOylation consensus site (in the case of stabilization) (Sampson *et al*, 2001), are in the vicinity of the phosphorylation site, which suggest potential cross-talk between different PTMs.

We envision that our large data set, which we have made freely available through an interactive website at http://phosphonetworks.org, will serve as a valuable resource for the research community in several ways. For instance, aside from generating more human KSRs and more phosphorylation motifs than all previous studies combined, this study also provides a blue print for mapping kinase-dependent connections and serves as a foundation for the development of new tools, such as genetically targetable kinase activity

reporters and phosphorylation site-specific antibodies, that promise to offer important insights into the regulation of kinases and their downstream substrates. Likewise, the large number of human KSRs identified in these studies can be used in conjunction with KSR data sets from other species (e.g., *S. cerevisiae*) to explore the extent to which kinase-dependent signaling networks are conserved across species (Hu *et al*, 2013). Moreover, in the future, it should be possible to integrate information from the human phosphorylation data set with that from other large-scale studies conducted on the proteome scale to gain a greater understanding of the organization and regulation of the signaling networks that govern cell physiology in human health and disease. Finally, in accord with recent initiatives put forth by the Human Proteome Organization (HUPO) (Paik *et al*, 2012), the CEASAR strategy developed in this study can be extended to construct signaling networks mediated by other post-translational modifications, such as ubiquitylation, SUMOylation, acetylation, and methylation, to gain global insights into a wide variety of signaling processes.

## Materials and methods

### Kinase purification

Two hundred and eighty-nine non-redundant human kinase genes obtained from the Invitrogen Ultimate Human ORF collection and other sources were cloned into the yeast expression vector, pEGH-A, using the Gateway cloning system (Invitrogen). Each clone was verified by restriction digestion. Each kinase was expressed as a GST fusion in the budding yeast, *Saccharomyces cerevisiae*, and purified using glutathione-sepharose affinity chromatography, as described previously (Zhu *et al*, 2001).

### Phosphorylation assays using protein microarrays

Human protein microarrays were generated as described previously (Hu *et al*, 2009). Each microarray contained 4191 unique proteins consisting of a collection of human proteins including TFs, RNA-binding proteins, DNA repair proteins, protein kinases and mitochondrial proteins as well as a panel of proteins involved in various other cellular processes. To identify *in vitro* substrates for each kinase, a protocol similar to that described by Zhu *et al* (2009), which involves radioactivity-based detection, was used.

### Bayesian approach

To predict KSRs that are likely to occur *in vivo*, we used a naïve Bayesian approach (Jansen *et al*, 2003; Hu *et al*, 2010) to integrate information about tissue-specific gene expression, subcellular localization, and PPIs. For the positive data set, we collected 1103 experimentally validated kinase-substrate pairs from the literature and the PhosphoELM database (phospho.elm.eu.org). We also constructed an artificial data set as a negative data set that contains 10 000 protein pairs where no kinases were included in the set. The relative weights for these three features were learned from the known data sets and applied to each of the rawKSRs.

### M3 algorithm

To predict consensus phosphorylation motifs for the kinases in our collection, we integrated the following data sources: the rawKSRs determined by protein microarray, the phosphorylation sites determined by MS/MS, and the phosphorylation sites with known upstream kinases obtained from the literature. Each site was mapped to the protein sequences of substrates identified in the rawKSR data set and subject to an iterative process by the M3 algorithm.

## Supplementary information

## Acknowledgements

We thank J Bader, JO Liu, M Li, J Beoke, and J Shen for their critical review and editing of the manuscript. We also thank C Whitehead, R Hanna, S Wei, and J de Melo for technical assistance and P Tucker and D Hayward for providing anti-ARID3A antisera and the pSG5-FLAG expression vector, respectively. This work was supported in part by the NIH grants (R01 DK073368 and DP1 CA174423 to JZ; RR020839 to AP; RR020839, DK082840, GM076102, CA125807, CA160036, and HG006434 to HZ; RR020839 to JQ; HG005220 and R01HG006282 to HJ; CA16519 to SD; EY019305 to DJZ; CA74305 to PAC; GM079498 to BET; and GM059802 to KND), Canadian Institutes for Health Research (MOP-84314 to A-CG), and the Structural Genomics Consortium (to SK).

*Author contributions:* RHN designed research, performed experiments, analyzed data, and wrote the paper; JH designed research, analyzed data, and wrote the paper; HSR designed research, performed experiments, analyzed data, and wrote the paper; XZ designed research, analyzed data, and wrote the paper; CW performed experiments and analyzed data; JN performed experiments and analyzed data; CC designed research and performed experiments; MS performed experiments; HC performed experiments; SH performed experiments; WH analyzed data; JSJ performed experiments; GW analyzed data; JL analyzed data; XG performed experiments; QN provided reagents or data sets; RG provided reagents or data sets; SX performed experiments; HJ provided reagents or data sets; KD provided reagents or data sets; MB provided reagents or data sets; PC provided reagents or data sets; SK provided reagent or data sets; AGR provided reagents or data sets; DJZ provided reagents or data sets; SB designed experiments; TP provided reagents or data sets; AG provided reagents or datasets; SD provided reagents or datasets; AP provided reagents or data sets; BET provided reagents or datasets; JZ designed research, analyzed data, and wrote the paper; HZ designed research, analyzed data and wrote the paper; JQ designed research, analyzed data, and wrote the paper.

## Conflict of interest

The authors declare that they have no conflict of interest.

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
