## [Review Process File · Molecular Systems Biology]

Construction of Human Activity-Based Phosphorylation Networks

Robert H. Newman, Jianfei Hu, Hee-Sool Rho, Zhi Xie, Crystal Woodard, John Neiswinger, Christopher Cooper, Matthew Shirley, Hillary M. Clark, Shaohui Hu, Woochang Hwang, Jun Seop Jeong, George Wu, Jimmy Lin, Xinxin Gao, Qiang Ni, Renu Goel, Shuli Xia, Hongkai Ji, Kevin N. Dalby, Morris J. Birnbaum, Philip A. Cole, Stefan Knapp, Alexey G. Ryazanov, Donald J. Zack, Seth Blackshaw, Tony Pawson, Anne-Claude Gingras, Stephen Desiderio, Akhilesh Pandey, Benjamin E. Turk, Jin Zhang, Heng Zhu, Jiang Qian

Corresponding author: Jiang Qian, Johns Hopkins

Review timeline:

Submission date:	17 August 2012
Editorial Decision:	20 September 2012
Revision received:	18 December 2012
Editorial Decision:	10 February 2013
Revision received:	22 February 2013
Accepted:	01 March 2013

Editors: Andrew Hufton / Thomas Lemberger

Transaction Report:

1st Editorial Decision

20 September 2012

Thank you again for submitting your work to Molecular Systems Biology. We have now heard back from the four referees who agreed to evaluate your manuscript. As you will see from the reports below, the referees, particularly the last three, were cautiously supportive. They raised, however, substantial concerns, which, I am afraid to say, must preclude its publication in its present form.

The reviewers provided detailed reports that raise a series of important technical and conceptual issues that will need to be convincingly addressed before this work would be acceptable for publication at Molecular Systems Biology. I highlight a few of these issues here:

1. More than one reviewer had issues with the quality of the presented gel data, and Reviewer #2 felt that the reproducibility of these results would need to be clearly demonstrated. Additional experiments and replicate analyses may be needed to address these concerns. In this case, we also ask that you provide the original unprocessed gel images for all gels presented in this work as supplemental material, in line with our policies on data release and scientific transparency. Please include the unprocessed/uncropped gel images as individual files within a single zip folder, and upload this as a supplemental dataset. Replicate gels should also be included when available. File names should clearly link the gels to the appropriate figure panels.

2. The reviewers found the phosphorylation-dependent changes in protein stability somewhat

surprising, and Reviewer #2 noted that all of the cases showing this pattern were from AGC family kinases, which raises issues with the generality of this observation. Additional discussion, and possibly further experiments with non-AGC kinases, may be needed.

3. We thank you for supplying much of the underlying data and results through your personal web-resource (<http://phosphonetworks.org>). To help ensure long term accessibility to published findings, we prefer to also incorporate key datasets into our supplementary materials whenever possible. As such, we ask that you provide the full high-resolution network (as requested by Reviewer #4), and the motif matrices in as supplementary files in a common machine-readable file format. If Cytoscape was used to generate Fig. 4A, we would also ask to provide the Cytoscape session file, to aid researchers that may wish to browse the entire network.

If you feel you can satisfactorily deal with these points and those listed by the referees, you may wish to submit a revised version of your manuscript. Please attach a covering letter giving details of the way in which you have handled each of the points raised by the referees. A revised manuscript will be once again subject to review and you probably understand that we can give you no guarantee at this stage that the eventual outcome will be favorable.

Referee reports:

Reviewer #1 (Remarks to the Author):

The title that this this manuscript bears is a bold statement, implying that the ensuing results may put to bed a central problem: assigning direct and unambiguous links between regulatory enzymes and their protein substrates. I can see that there is an enormous amount of valuable data in this manuscript, but what is presented falls short of solving the stated problem and the methods described do not represent an advance beyond ideas already well trodden. At the outset, a central premise of the approach the authors describe is that *in vitro* measurement of phosphorylation of a protein by a kinase can somehow be linked to the existence of observable phosphorylation of specific peptides *in vivo* as determined by mass spectroscopy. The former was pioneered by Ptachek, et al. (Science, 2005) to a questionable degree of success, as anyone who has tried to use this data in followup work will attest. Equating *in vivo* interactions of kinase and phosphatase with putative substrate, Breitkreutz, et al., (Science 2010) claimed to relate observed interactions to existence of phosphorylation of the substrates measured by MS. It was not clear in that case, what relationships could be drawn at all. A fundamental flaw with this manuscript and all preceding examples that claim to establish direct kinase-substrate relationships is that it is now clear how much of the phosphorylation that is observed *in vivo* has nay functional meaning (Lienhard, TiBs 2008). The filtering strategy the authors describe cannot address this issue. One key piece of information that help is to ask what is the proportion of a protein in the cell that is phosphorylated, as suggested by Wu, et al. (Nat. Methods, 2011) and Levy, et al., (Phil. Trans. B, 2012). This would require that the abundances of protein under which a phosphoproteomic analysis was performed be measured. Just to give an example. Imagine that you observe strong signal for a kinase on a particular substrate *in vitro*. You then check the MS phosphproeomic data and sure enough, it is phosphorylated *in vivo*. That's great. Now you overexpress the kinase and viola! Your substrate is phosphorylated *in vivo*! But here is the problem. In both cases the phosphorylation could be nonspecific. You can't know, though again, a reasonable assumption would be that if the stoichiometry of phosphorylation *in vivo* were high, and the site that is phosphorylated is evolutionarily conserved, you might have a leg to stand on. But again, in the absense of abundance data I do not believe you can say much. Nor do I buy the other types of inference and biological validation presented in this manuscript. It does not surprise me that if a protein can be phosphorylated by a kinase, that if you overexpress that kinase then stability of the protein could be affected. Finally, I don't buy that the any enrichment of motifs among substrates carries any special meaning. I would want to see controls The examples given are of those rare cases of kinases that are already well know to have clearly defined motifs (e.g. PKA, Cdk, MAPKs). The authors know that these are exceptions, not the rule.

Having hammered away at this manuscript I must admit that I do think the work is quite valuable and could be of value to the signaling community; just not as claimed to be a high resolution anything. Perhaps if written in a more circumspect manner and taking into account all caveats it

could prove, as the authors put it, "a valuable resource to the research community".

Reviewer #2 (Remarks to the Author):

Newman and collaborators developed a new strategy to build high-resolution phosphorylation networks in humans in which each kinase is associated with one or more specific phosphoproteins and phosphosites on these phosphoproteins. The strategy employs protein microarrays, phosphoproteomics data and computational data integration. The authors first apply this strategy to build a network of 230 human kinases, 652 substrates and 2,500 phosphosites. The authors then focus on one specific sub-network and study the B cell receptor signaling pathway, unraveling a new link between Btk and PKA as well as a new functional role for the PKA in this pathway. The large-scale experiments, computational analyses and small-scale confirmation experiments are sound in general.

I consider this work to be of great interest for the readership of Molecular Systems Biology because:

1. to my knowledge this is the first effort to build high-resolution kinase-phosphosite networks in humans and at the level of phosphorylation sites in any species.
2. this approach represents a clear and nice example of integration of different experimental data types and computational predictions.
3. the data presented here will undoubtedly serve as seeds for many more studies to come on the architecture and the dynamics of signaling networks in human cells.

Although I would tend to be highly supportive of the publication of this manuscript, I have some concerns that in my opinion should be considered before it can be published in Molecular Systems Biology.

Major concerns

1. The authors show that the changes in protein stability they observed are due to phosphorylation of the substrate by its cognate kinase. They select 20 KSRs in which the kinases are PKC, PKA, PKB and Erk2. The first three kinases belong to the same group (even if they are part of different families), the AGC group, while Erk1 belongs to the CMGC group. The authors succeed in showing that the changes in protein stability they observed are due to phosphorylation of the substrate by its cognate kinase for 16 out of 20 KSRs. All the positive KSR are for kinases belonging to the AGC group. The authors should discuss if this behavior is general or specific to the AGC group.
2. In general, the signal of the loading controls on the blots is weak and difficult to use as reference. This can lead to misinterpretations of the results. For instance, the results shown in Figure 2C appear not to match the description provided in the main text. In the top panel, the decrease of the abundance of HCLS is not clear since the L.C. signal is too weak. In the bottom panel, while the PKA substrates are generally phosphorylated after 30', nothing definitive can be assessed about HCLS phosphorylation as proposed in the main text. Moreover, the abundance of HCLS does not seem to decrease as shown in the top panel. The addition of H89, which is supposed to block PKA activity, seems to decrease HCLS abundance, whereas the opposite was expected.
3. It is not clear from figure 2A what is the observed effect for the first case, HCLS1 and NEK10. Do we really see a different mobility or a difference in abundance? Given the lack of clarity of some of the results presented in this figure, results of all of the 243 KSRs assayed should be presented in supplementary material, not just the summary statistics.
4. Figure 2B bottom. What is the antibody that recognizes phosphorylated AKT substrates? What is being measured here exactly? The legend of figure 2 is rather incomplete and referring to the main text is not satisfying.
5. Figure 3. Again, the figure should be explained in the legend and should not refer to the main text or supp. material.
6. Figure 4B. In general the blots are of poor quality and the signal is extremely weak and L.C. barely visible. The reproducibility of these assays should be assessed and shown. Also, in the case of SMAD4-CDK7, protein abundance decreases when the kinase is present in the wild-type but increases in the mutant when the kinase is present. How could that be explained?
7. The results shown in Supplementary Figure 11C are not convincing since in the second column the increase in PKA substrates phosphorylation seems to follow the increase in actin abundance.
8. In general, I would suggest the authors to provide a more detailed statistical analysis of the reliability of the network produced (false negative and false positive rates) that would be a valuable

reference for the future studies. Few associations kinase-phosphosite were confirmed experimentally but we would like to have a measure of confidence on the entire network.

Minor concerns

1. p.4 In the list of citations "(Breitkreutz et al., 2010; Fiedler et al., 2009; van Wageningen et al., ; Linding et al., 2007)", no year is specified for Wageningen et al.
2. p.4 I would suggest the authors to remove the list of citations from the sentence "...have uncovered important clues about the organization and regulation of kinase-mediated signaling pathways (Breitkreutz et al., 2010; Fiedler et al., 2009; van Wageningen et al., ; Linding et al., 2007),...". They have already cited these works before and it seems clear to me that the sentence refers to the same works.
3. p.6 the authors should mention how many proteins were removed from the analysis because of autophosphorylation.
4. p.8 "...the most common change observed was altered protein levels, suggesting that many kinases may control protein stability, either directly or indirectly". I would like the authors to support this sentence by providing some numbers. Further, it was surprising to me to see no reference associated with this observation. I did not expect that the most frequent changes would be changes in protein levels. The authors need to discuss if this is expected, if yes why, and if not why and provide references reporting similar observations.
5. p.15 "First, we observed that Tyr phosphorylation of PKA, which increased ~2-fold ten minutes after BCR activation, was prevented by pre-treatment of the B cells with the Btk-specific inhibitor, terric acid (TA) (Figure 5H). This correlated with an increase in the extent of PKA-mediated phosphorylation on a portion of cellular PKA substrates, including ARID3A (Figure 5I,J)". It is not clear to which part of the first sentence "This" refers to.
6. There is no reference regarding terric acid being a Btk inhibitor. Is this a specific one? Need support here.
7. The normalization process is not sufficiently detailed in the Supplementary Material.
8. Sometimes the authors use the acronym HCLS for Holocarboxylase synthetase, instead of HLCS. A striking example is present on p. 9. The same thing is applicable to Ks-KSI in Supplementary Figure 6-7.
9. Supp. Material. No page number.
10. Supp. Material, *Saccharomyces cerevisiae* is misspelled.
11. Supp. Material. Gelman et al. reference is incomplete.
12. It is not clear what the negative set constructed with 10,000 protein pairs is. Is it justifiable to use non-kinase proteins in this case? Several types of negative sets should be considered to see what the effects would be.
13. Supp. Material. Statistics about co-expression, colocalization etc. are not reported in the section on Implementation. The authors only mention significant differences or enrichments but numbers need to be provided.

Reviewer #3 (Remarks to the Author):

The study of Zhang, Zhu and Qian and coworkers describes a very comprehensive screen and follow up of the substrates for 289 human kinases encompassing over 2,656 kinase substrate relationships. The expression, KSRs, analysis pipeline, and follow up with the Btk, PKA pathway relationships represent a very useful new analysis method for defining kinase pathways.

The approach to compare the in vitro KSRs with the published shotgun MS/MS data is a clever idea for large-scale phospho-motif determination. There is an implicit assumption that the phosphorylation detected in vitro holds true in vivo. Can the authors comment on the issue of whether proteins identified as phosphorylated on the arrays by the presence of radioactive labeling matches the sites identified in the shotgun phosphoproteomic databases? This needs clarification because the way it reads now the authors suggest the sites are the same. If the authors could clarify this issue in the text with their current data it would strengthen the manuscript.

It is interesting that in the majority of cases the authors found changes in protein stability in the co-transfection experiments. I am wondering what the reason for this could be. Are the majority of phosphosites in phosphodegrons? Or does overexpression of important kinases like Akt, MAPK, etc.

have global effects on the cell that results in changes in the protein expression pattern. Are known oncogenic kinases for example more likely to generate changes in the expression pattern in their system? I think the authors could comment on this issue.

Once these textual issues are addressed I suggest the manuscript be accepted for publication. The authors' have done a remarkable job in assembling a map of phosphorylation sites which will be rapidly utilized by others in the field.

Reviewer #4 (Remarks to the Author):

In their article "Construction of a High-Resolution Map of Phosphorylation Networks in Humans" Newman et al. present an extensive, experimentally derived kinase-substrate network, conduct an extensive experimental validation, and provide an example of its value for the further elucidation of a well-studied signaling pathway.

With their study, the authors address a pertinent issue for the global analysis of signaling networks. Recent phospho-proteomics methods can measure approximately 20,000 distinct phosphorylation events in cells. However, only for a small minority of these sites the upstream kinases are known, which makes the reconstruction of phosphorylation networks challenging. The presented data set significantly contributes to fill this knowledge gap by providing 3,656 high-quality kinase-substrate relationships (KSRs).

Other studies have addressed this issue, previously. For example, Linding et al. 2007 (NetworKIN) and Miller et al. 2008 (NetPhorest) have predicted kinase-substrate relationships using sequence-based classifiers alone or by also considering the network context. In contrast to these studies, the current study is based on a more extensive experimentally derived data set using protein-array kinase assays.

Highlights of this study

- The authors present the currently most comprehensive, experimentally-derived data set of kinase-substrate relationships
- The authors conducted extensive validation experiments to demonstrate the quality of the derived KSRs
- The authors devised a novel algorithm (M3) that identifies position weight matrices from mixed (but enriched) target site populations
- The authors use a clever strategy (including their novel M3 algorithm) to identify specific phosphorylation sites from a dataset that (by itself) only resolves general kinase-substrate relationships

Caveats to keep in mind

- The tested substrate proteins were selected for a different purpose (as DNA-binding proteins). The potential bias due to this selection should be accounted for, when the data set is used in future analyses
- The approach has a relatively high false negative rate of 95% as judged by the recovery of kinase-substrate relationships from a literature set.

Overall, this study presents a major contribution to our understanding of phosphorylation networks. The KSR data set is extensive and the authors convincingly demonstrate its high quality. Beyond the presented experiments, the KSR data set represents an important resource for future systems-level analyses. With this, I would in principle recommend this manuscript for publication in MSB. However, first I would ask the authors to address the following issues:

Main text

- Page 6 "Individual protein microarrays, composed of 4,191 unique, full-length human proteins from 12 major protein families(Hu et al., 2009)": The use of protein rather than peptide arrays to measure kinase specificities/activities can be regarded as a major plus of this study - kinase activities are restrained by the substrate accessibility in the folded protein structure and additional protein interactions can play a role. However, this potential benefit is neither analyzed nor discussed. Include a discussion on this topic and analyze the potential benefit of protein arrays for this study. Also include a discussion of potential deficiencies, e.g. effects of improper folding or artificial steric

hinderance.

- Page 6 "from 12 major protein families (Hu et al., 2009)": The functional group targeted in the original publication was "DNA binding proteins". Thus, the protein list is highly biased toward this protein class. Such a bias could lead to analysis artifacts in subsequent (global) studies using this dataset. To prevent these potential problems, the authors need to more clearly communicate the extent and potential implications of this bias. Extend the discussion on the selection of these proteins, show the distribution of protein classes (and subcellular localization) in comparison with the whole proteome in a main figure, and analyze, whether there is any bias in the recovery of known KSRs comparing the selected protein set with the remaining proteins.

- Page 7 "Using a P value of 0.05 as a threshold...": the "p value threshold" was estimated based on a permutation of L values of the three selected features. Explain the rationale of choosing this p-value estimation approach.

- Page 7 "Second, cross-validation analysis revealed that the Bayesian approach increases the recovery rate of known KSRs over 5-fold": this analysis also suggests a high false negative rate of 95% (only 22 of 421 known KSRs were identified). This should be more clearly stated in the manuscript.

- Page 8 "the most common change observed was altered protein levels": this observation is really striking and its discussion should be extended, e.g. by discussing the potential mechanistic basis and by comparing the properties of these phosphorylation sites with sites that are known to affect protein stability.

- Page 10 "This approach combines our rawKSR data...": Discuss the rationale of selecting the rawKSR rather than the refKSR or comKSR dataset

- Page 11 "To independently validate the identified phosphorylation motifs, we compared our predicted phosphorylation motifs for 24 kinases to those obtained using a positional scanning peptide library": Extend this comparison to the "Linear Motif Atlas" published by Miller et al. (2008) (<http://netphorest.info>).

- Page 12 "what effect mutation of the predicted phospho-acceptor site had on the substrates in the presence of kinase for four selected KSRs.": Explain, how these four KSRs were selected. Were additional KSRs tested?

- Page 12 "A high-resolution map of human phosphorylation networks": Compare these results with the results from the NetworkKIN approach (http://networkin.info/version_2_0/search.php). For this, also separately evaluate the 768 known and the 3659 novel KSRs. This will allow the reader to better appreciate the benefits of the presented vs. previously published approaches.

Figures

- Fig. 1: Add a short summary legend.

- Fig. 2A: The "mobility shift" observed by the authors for the two selected KSRs is not (clearly) visible. This is especially the case for the first pair (HCLS1/NEK10), for which the substrate band is overexposed. For the 2nd pair (PRKACA/EPHA10) a decrease of the lower band is visible, but it is unclear how reproducible this (minor) effect is. While the (de-)stabilization effects are more convincing, the authors need to provide more evidence for the observed "mobility shifts". The authors might consider Phos-tag acrylamide (<http://www.phos-tag.com/english/>) for more pronounced shifts.

- Fig. 4B: For some of the tested pairs the effect of substrate mutation is not apparent. For example, the blots for the CBL-S900/MAPK15 pair still show an increase in substrate levels. Overall, it should be made clear, how reproducible these results were. In addition, for the PRKAR1A/PRKACA pair substrate mutation affects the protein level by itself. This should be explained.

Supplement

- Include the high-resolution network as a text or excel file (instead of a pdf file)

- Include the actual position weight matrices (PWM) derived from the M3 algorithm in addition to their graphical representations

- Include figure of all Western Blot results from the first-stage validation studies

- Page 8 "To this end, we collected 1,108 known KSRs from database and literature sources." 1,103 "known KSRs" were used before as the positive training set. Why this difference in the reported numbers?

- Fig. S4.: explain F/B ratio in legend

Thank you very much for considering our manuscript for publication in *Molecular Systems Biology* and for giving us the opportunity to address the concerns raised by the Reviewers. The Reviewers' comments were very insightful and, we believe, served to improve the overall quality of the manuscript. We believe that we have addressed all of their concerns in the accompanying response letter in a point-by-point fashion and revised the manuscript accordingly. We also wanted to be sure that we satisfactorily addressed the points that you raised in the decision letter. Please see our responses below.

- 1. More than one reviewer had issues with the quality of the presented gel data, and Reviewer #2 felt that the reproducibility of these results would need to be clearly demonstrated. Additional experiments and replicate analyses may be needed to address these concerns. In this case, we also ask that you provide the original unprocessed gel images for all gels presented in this work as supplemental material, in line with our policies on data release and scientific transparency. Please include the unprocessed/uncropped gel images as individual files within a single zip folder, and upload this as a supplemental dataset. Replicate gels should also be included when available. File names should clearly link the gels to the appropriate figure panels.**

With regard to the quality of the presented gel data, we have re-scanned many of the original films using a translucent (film) scan function. We believe that this step substantially improves the image quality of the presented data. We have also included replicate gels and quantitated band intensities for many of the images. In addition, we have compiled a spreadsheet with links that allow the user to quickly and easily access all of the unprocessed data images from the validation experiments. As you requested, we have submitted the zip folder containing these raw data as a supplemental dataset.

- 2. The reviewers found the phosphorylation-dependent changes in protein stability somewhat surprising, and Reviewer #2 noted that all of the cases showing this pattern where from AGC family kinases, which raises issues with the generality of this observation. Additional discussion, and possibly further experiments with non-AGC kinases, may be needed.**

As discussed in much greater detail in our responses to Reviewer 2 (major "concern #1" and "minor concern #4") and Reviewer 3 ("major concern #2"), we believe that the frequency with which we observe kinase-dependent changes in substrate levels can be attributed, at least in part, to the fact that many of the substrates in our test set are transcription factors (TFs). Indeed, phosphorylation-dependent modulation of protein levels appears to be a common mode of regulation among TFs (as well as other protein families). Moreover, for various technical reasons (e.g., the negatively charged FLAG tag (DYKDDDDK) used for detection is likely to mask phosphorylation-dependent mobility shifts in many substrates), it is possible that some kinase-dependent changes are under-represented in our dataset.

With regard to the question about the generalizability of kinase-dependent changes in protein stability to other kinase groups, as outlined in our response to Reviewer 3 ("major concern #2"), this phenotype was observed for substrates of kinases from several different kinase groups (e.g., AGC, CGMC, NRTK, etc.), suggesting that it is not specific to the AGC group. We were, however, somewhat limited in the KSR pairs that we could examine in the second stage validation experiments (i.e., phospho-substrate probing) due to the availability of commercial phospho-substrate specific antibodies and specific pharmacological activators/inhibitors of kinase activity. As a consequence, many of the kinases that we examined in this round of experiments were from the AGC group.

- 3. We thank you for supplying much of the underlying data and results through your personal web-resource (<http://phosponetworks.org>). To help ensure long term accessibility to published findings, we prefer to also incorporate key datasets into our supplementary materials whenever possible. As such, we ask that you provide the full high-resolution network (as requested by Reviewer #4), and the motif matrices in as**

supplementary files in a common machine-readable file format. If Cytoscape was used to generate Fig. 4A, we would also ask to provide the Cytoscape session file, to aid researchers that may wish to browse the entire network.

This is a great suggestion. We have included all of the datasets as supplementary material.

Reviewer #1 (Remarks to the Author):

The title that this this manuscript bears is a bold statement, implying that the ensuing results may put to bed a central problem: assigning direct and unambiguous links between regulatory enzymes and their protein substrates. I can see that there is an enormous amount of valuable data in this manuscript, but what is presented falls short of solving the stated problem and the methods described do not represent an advance beyond ideas already well trodden. At the outset, a central premise of the approach the authors describe is that in vitro measurement of phosphorylation of a protein by a kinase can somehow be linked to the existence of observable phosphorylation of specific peptides in vivo as determined by mass spectroscopy. The former was pioneered by Ptacek, et al. (Science, 2005) to a questionable degree of success, as anyone who has tried to use this data in followup work will attest. Equating in vivo interactions of kinase and phosphatase with putitive substrate, Breitkreutz, et al., (Science 2010) claimed to relate observed interactions to existence of phosphorylation of the substrates measured by MS. It was not clear in that case, what relationships could be drawn at all. A fundamental flaw with this manuscript and all preceding examples that claim to establish direct kinase-substrate relationships is that it is now clear how much of the phosphorylation that is observed in vivo has nay functional meaning (Lienhard, TiBs 2008). The filtering strategy the authors describe cannot address this issue. One key piece of information that help is to ask what is the proportion of a protein in the cell that is phosphorylated, as suggested by Wu, et al. (Nat. Methods, 2011) and Levy, et al., (Phil. Trans. B, 2012). This would require that the abundances of protein under which a phosphoproteomic analysis was performed be measured. Just to give an example. Imagine that you observe strong signal for a kinase on a particular substrate in vitro. You then check the MS phosphproeomic data and sure enough, it is phosphorylated in vivo. That's great. Now you overexpress the kinase and viola! Your substrate is phosphorylated in vivo! But here is the problem. In both cases the phosphorylation could be nonspecific. You can't know, though again, a reasonable assumption would be that if the stoichiometry of phosphorylation in vivo were high, and the site that is phosphorylated is evolutionarily conserved, you might have a leg to stand on. But again, in the absense of abundance data I do not believe you can say much. Nor do I buy the other types of inference and biological validation presented in this manuscript. It does not surprise me that if a protein can be phosphorylated by a kinase, that if you overexpress that kinase then stability of the protein could be affected. Finally, I don't buy that the any enrichment of motifs among substrates carries any special meaning. I would want to see controls. The examples given are of those rare cases of kinases that are already well know to have clearly defined motifs (e.g. PKA, Cdks, MAPKs). The authors know that these are exceptions, not the rule.

Having hammered away at this manuscript I must admit that I do think the work is quite valuable and could be of value to the signaling community; just not as claimed to be a high resolution anything. Perhaps if written in a more circumspect manner and taking into account all caveats it could prove, as the authors put it, " a valuable resource to the research community".

RESPONSE:

Thanks to the reviewer for your insightful comments. We are aware of the challenges of constructing phosphorylation networks in higher eukaryotes. Foremost, determination of kinase-substrate relationships (KSRs) requires activity-based assays. To our knowledge, this is the first high-throughput study to identify human KSRs based on direct, biochemical reactions. This distinguishes our study from those using other high-throughput technologies, such as yeast two-hybrid and TAP-tagged approaches, synthetic genetic screening, and shotgun MS/MS. For instance, although various shotgun MS/MS approaches have generated an extremely large number of phosphorylated sites in mammals, these approaches cannot connect the identified sites to their immediate upstream kinases, namely direct KSRs. Our study combines in vitro protein microarray-based assays, bioinformatics analysis, and thorough validation studies to narrow this large knowledge gap (i.e., direct KSRs) in human phosphorylation networks.

Secondly, we agree that kinase-substrate relationships determined *in vitro* do not necessarily reflect the *in vivo* relationships. However, in contrast to Ptacek's work, here we employed a series of bioinformatics approaches, such as Bayesian analysis and data integration, to *enrich for* physiologically relevant phosphorylation events. This led to the generation of the refKSR dataset. Aside from computational evidence, the utility of these analyses was further strengthened by a series of unbiased, cell-based validation experiments, a standard approach employed by the signaling community.

- a) In the first round of cell-based experiments, the 243 KSR pairs examined included 75 unique kinases and 136 substrates, which were randomly selected (Page 9).
- b) During the second round of validation experiments, we focused on a subset of KSRs. While the substrates used in these analyses were again randomly selected, the kinases were not, due to the limitations imposed by the availability of commercially available drugs and phosphorylation-specific antibodies (Page 10).
- c) Furthermore, when site-directed mutagenesis assays were performed, random KSRs were again chosen (Page 14).
- d) Finally, the experiments used to validate both the Btk-PKA and the PKA-ARID3A KSRs were conducted in untransfected human B cells, where each of the proteins is expressed at endogenous levels (Page 15-17).

In each case, we observed a high success rate, validating the refKSRs.

In summary, to ensure that the refKSRs are enriched to reflect physiologically relevant events, we thoroughly tested our predictions using a series of *in vitro* and cell-based approaches, including 1) co-expression of KSR pairs in cells, 2) activation/inhibition of endogenous kinases using both agonists (e.g., PMA, forskolin) and antagonists (e.g., H89, TA) coupled with detection of kinase-specific phosphorylation events using phosphorylation-specific antibodies, and 3) site-directed mutagenesis involving both cell-based assays and *in vitro* phosphorylation reactions. The high rates of success observed in all of the above validation studies strongly suggest that the refKSRs are of high fidelity.

In response to the reviewer's comments, we have tried to soften some of the claims made in the manuscript and agree to change the title to "Construction of activity-based phosphorylation networks in humans".

Reviewer #2 (Remarks to the Author):

Newman and collaborators developed a new strategy to build high-resolution phosphorylation networks in humans in which each kinase is associated with one or more specific phosphoproteins and phosphosites on these phosphoproteins. The strategy employs protein microarrays, phosphoproteomics data and computational data integration. The authors first apply this strategy to build a network of 230 human kinases, 652 substrates and 2,500 phosphosites. The authors then focus on one specific sub-network and study the B cell receptor signaling pathway, unraveling a new link between Btk and PKA as well as a new functional role for the PKA in this pathway. The large-scale experiments, computational analyses and small-scale confirmation experiments are sound in general.

I consider this work to be of great interest for the readership of Molecular Systems Biology because:

- 1. to my knowledge this is the first effort to build high-resolution kinase-phosphosite networks in humans and at the level of phosphorylation sites in any species.*
- 2. this approach represents a clear and nice example of integration of different experimental data types and computational predictions.*
- 3. the data presented here will undoubtedly serve as seeds for many more studies to come on the architecture and the dynamics of signaling networks in human cells.*

Although I would tend to be highly supportive of the publication of this manuscript, I have some concerns that in my opinion should be considered before it can be published in Molecular Systems Biology.

Major concerns

1. The authors show that the changes in protein stability they observed are due to phosphorylation of the substrate by its cognate kinase. They select 20 KSRs in which the kinases are PKC, PKA, PKB and Erk2. The first three kinases belong to the same group (even if they are part of different families), the AGC group, while Erk1 belongs to the CMGC group. The authors succeed to showing that the changes in protein stability they observed are due to phosphorylation of the substrate by its cognate kinase for 16 out of 20 KSRs. All the positive KSR are for kinases belonging to the AGC group. The authors should discuss if this behavior is general or specific to the AGC group.

RESPONSE: We would like to thank the reviewer for the positive comments. Regarding the generalizability of the observed changes in protein stability as a result of phosphorylation, we do not believe that this phenomenon is only applicable to the AGC group. First, it has been reported in the literature that phosphorylation by kinases of other groups can change protein stability as well. For example, Cyclin E/Cdk2 (CMGC group)-mediated phosphorylation promotes the stabilization of the replication initiation factor, Cdc6, during the transition of quiescent cells back into the cell cycle (Mailand and Diffley, *Cell* (2005) 122, 915). Secondly, in our first round of validation, we randomly selected 243 KSRs, including 75 unique kinases and 136 substrates. For those pairs with altered stability, the kinases span several kinase groups, including AGC (e.g., PKA, PKC, and Akt), CGMC (e.g., MAPK1, MAPK8, CDC2, and CDK4), and NRTK (e.g., TYK2, FYN, and PTK2B). In each case, we observed both stabilization and degradation phenotypes. In fact, overexpression of the same kinase often led to differential changes on the stability/electrophoretic mobility of different substrates (e.g., overexpression of STK17A led to *i*) a reduction in the levels of FIP1L1, ELL3, and PCBP2, *ii*) an increase in the levels of ASCC1, ENO1 and WHSC2, and *iii*) no change in the levels of ARID3A, IRF5, and MEOX1), suggesting the observed changes are not due to global changes in cell proliferation. The reason that we focused on kinases in the AGC group during our second round of validation is that we were limited to those kinases for which both specific activators/inhibitors and phosphorylation-specific antibodies were commercially available. Unfortunately, this combination of reagents is not readily available for kinase families other than AGC and CMGC family members. We added a discussion on this issue (Page 20).

2. In general, the signal of the loading controls on the blots is weak and difficult to use as reference. This can lead to misinterpretations of the results. For instance, the results shown in Figure 2C appear not to match the description provided in the main text. In the top panel, the decrease of the abundance of HCLS is not clear since the L.C. signal is too weak. In the bottom panel, while the PKA substrates are generally phosphorylated after 30', nothing definitive can be assessed about HCLS phosphorylation as proposed in the main text. Moreover, the abundance of HCLS does not seem to decrease as shown in the top panel. The addition of H89, which is supposed to block PKA activity, seems to decrease HCLS abundance, whereas the opposite was expected.

RESPONSE: Regarding the loading control in the top panel of Figure 2C, we typically observed approximately three non-specific bands in the FLAG blots when using HeLa cell lysates. Importantly, the levels of these non-specific bands remained constant in the presence or absence of kinase and pilot experiments demonstrated that changes in their levels mirrored the levels observed for tubulin in the same blots. When tubulin or actin was not used as the loading control, one of these bands was used instead. However, in this particular case, HLCS migrates close to the non-specific band used as the loading control, making it difficult to crop the image. To remedy this, we have substituted an image showing both HLCS and the non-specific band (each is marked). We have also quantitated the band intensities in panels B and C. The normalized intensity ratio (substrate intensity/loading control intensity) for each band is shown below the corresponding lane (Page 10).

With respect to the bottom panel of Figure 2C, note that these experiments were: 1) conducted over a shorter time course than the first round validation experiments (<4 h of endogenous PKA activation in the bottom panel vs 16h overexpression of PKA in the top panel) so we would not necessarily expect to see the same degree of stabilization as observed in the first stage experiments, and 2) they involved immunoprecipitation of FLAG-tagged HLCS. Because the IP can either 1) mask reductions in protein levels (e.g. if the antibody binding capacity is exceeded) or 2) result in reduced yield due to poor binding or experimental variation, it is best to compare the levels

of PKA-mediated phosphorylation at the various time points to the amount of total FLAG-HLCS in that lane (i.e. the FLAG blot). Therefore, as mentioned above, we have quantitated the signal intensities in the bottom panel of Figure 2C. The normalized intensity ratios demonstrate that there is both an increase in PKA-mediated phosphorylation of HLCS following activation of endogenous PKA and a reduction in the extent of PKA-mediated phosphorylation on HLCS in the presence of H89. Note that, though H89 does not completely abolish phosphorylation by PKA after 4 h of Fsk treatment, there is an approximately 3-fold reduction. In our experience, after 4.5 h, the potency of H89-mediated inhibition is often reduced.

3. It is not clear from figure 2A what is the observed effect for the first case, HCLS1 and NEK10. Do we really see a different mobility or a difference in abundance? Given the lack of clarity of some of the results presented in this figure, results of all of the 243 KSRs assayed should be presented in supplementary material, not just the summary statistics.

RESPONSE: All the gel images for the 243 KSRs are included as supplementary materials. Moreover, the HLCS1/NEK10 image has been replaced with another example of a kinase-dependent mobility shift, CUEDC2/PRKACA, that, we hope, is clearer.

4. Figure 2B bottom. What is the antibody that recognizes phosphorylated AKT substrates? What is being measured here exactly? The legend of figure 2 is rather incomplete and referring to the main text is not satisfying.

RESPONSE: The antibody, anti-S/T-phospho-Akt substrate antibody, is a commercially-available, polyclonal antibody raised against phosphorylated Akt substrates based on the consensus motif: K/RXK/RXXS/TX (Cell signaling technology, Cat. No. 9611). In the bottom panel of Figure 2B, FLAG-ZRANB2 was immunoprecipitated at the indicated times (0 and 20 min) after stimulation and probed for Akt-mediated phosphorylation (using the anti-phospho-Akt substrate antibody described above). The blot was then stripped and re-probed for total FLAG-ZRANB2 using an anti-FLAG antibody. Despite a large reduction in total FLAG-ZRANB2 after 20 minutes of Akt stimulation, a large portion of the protein is phosphorylated.

That being said, in retrospect, we feel that the marked reduction in Akt levels following insulin treatment makes this particular example less intuitive than other examples in this dataset. Therefore, we have replaced the ZRANB2/Akt example with another example, specifically DDEF1/PKC. Another example of the ZRANB2/Akt interaction, in which the ZRANB2 levels did not change as dramatically, is shown in the summary figure (Page 10).

5. Figure 3. Again, the figure should be explained in the legend and should not refer to the main text or supp. material.

RESPONSE: We added more details in the figure legend (Page 27).

6. Figure 4B. In general the blots are of poor quality and the signal is extremely weak and L.C. barely visible. The reproducibility of these assays should be assessed and shown. Also, in the case of SMAD4-CDK7, protein abundance decreases when the kinase is present in the wild-type but increases in the mutant when the kinase is present. How could that be explained?

RESPONSE: These blots have been re-scanned using a “film scan” function to improve the signal quality (the unprocessed, re-scanned images are included in the revised Supplemental Materials Section, as well). The quantitation shown below each blot in Figure 4 now represents the average of at least two independent experiments. The standard error for each is shown.

With respect to the SMAD4-CDK7 set, the differential effects of CDK7 on WT and MT SMAD4 levels (i.e., degradation in the case of WT and stabilization in the case of MT) are suggestive of a complex mode of SMAD4 regulation by CDK7. This curious phenotype was the reason that we originally did not consider the SMAD4-CDK7 set as a validated prediction. We have repeated these experiments several times (>5 times) in the same cell line (HeLa), as well as in HEK293T cells. In most cases we observe a similar phenotype, but not always. Therefore, since the underlying biological mechanism is not currently clear and warrants more investigation, we have

removed the SMAD4-CDK7 set from the figure. We believe that the removal of this set does not affect our overall conclusions.

7. *The results shown in Supplementary Figure 11C are not convincing since in the second column the increase in PKA substrates phosphorylation seems to follow the increase in actin abundance.*

RESPONSE: We have quantitated the total phospho-PKA intensity (i.e. the signal in the entire lane) and compared it to the intensity of the actin loading control. We see an ~2.4-fold increase in normalized p-PKA signal upon Btk overexpression. In contrast, the normalized p-PKA signal is reduced to 50% of the reference in the presence of H89. We have revised the figure to reflect this. The results are shown graphically below:

8. *In general, I would suggest the authors to provide a more detailed statistical analysis of the reliability of the network produced (false negative and false positive rates) that would be a valuable reference for the future studies. Few associations kinase-phosphosite were confirmed experimentally but we would like to have a measure of confidence on the entire network.*

RESPONSE: 758 of 1156 known site-specific kinase-substrate interactions were recovered by our method. Therefore, the false negative rate is $1-758/1156=34.4\%$. Based on our validation experiments, each of the 5 kinase-phosphosite interactions tested (including the Btk→PKA and PKA→ARID3A sets) were confirmed by protein mutagenesis experiments. Therefore, the false positive rate appears to be very low. (Page 19).

Minor concerns

1. *p.4 In the list of citations "(Breitkreutz et al., 2010; Fiedler et al., 2009; van Wageningen et al., ; Linding et al., 2007)", no year is specified for Wageningen et al.*

RESPONSE: Done (Page 5).

2. *p.4 I would suggest the authors to remove the list of citations from the sentence "...have uncovered important clues about the organization and regulation of kinase-mediated signaling pathways (Breitkreutz et al., 2010; Fiedler et al., 2009; van Wageningen et al., ; Linding et al., 2007),...". They have already cited these works before and it seems clear to me that the sentence refers to the same works.*

RESPONSE: Done (Page 5).

3. *p.6 the authors should mention how many proteins were removed from the analysis because of autophosphorylation.*

RESPONSE: We removed 17 proteins due to autophosphorylation and 35 additional proteins due to high binding affinity to ATP in the negative control experiment (Page 7).

4. *p.8 "...the most common change observed was altered protein levels, suggesting that many kinases may control protein stability, either directly or indirectly". I would like the authors to*

support this sentence by providing some numbers. Further, it was surprising to me to see no reference associated with this observation. I did not expect that the most frequent changes would be changes in protein levels. The authors need to discuss if this is expected, if yes why, and if not why and provide references reporting similar observations.

RESPONSE: Overall, 112 of the 132 (~85%) substrates that exhibited kinase-dependent changes in the first round validation studies were either stabilized (51/132; 39%) or destabilized (61/132; 46%) in the presence of their cognate kinase. While this result was not necessarily expected, in retrospect, it is not surprising for the following reasons:

- 1) Phosphorylation-dependent modulation of protein levels appears to be a common mode of regulation among transcription factors (TFs). Indeed, a cursory search of the literature identified many instances in which the protein levels of TFs (and other DNA-binding proteins) are modulated by phosphorylation (see “citation list 1A” below). TFs were over-represented in our refKSR dataset due to the nature of the microarrays used during our *trans*-phosphorylation assays (for a description of the microarrays used in these experiments, please see Hu et al, *Cell* **139**, 610-622 (2009)). Therefore, it is not necessarily surprising that a large number of the substrates in our first round validation studies underwent kinase-dependent changes in their protein levels. In fact, ~54% of the substrates that underwent kinase-dependent changes in their stability were TFs, despite the fact that only ~35% of the proteins in the test set were found within this family. Of course, this mode of regulation is not limited to TFs, as evidenced by several instances in the literature in which members of other protein families also have been reported to undergo phosphorylation-dependent changes in their protein levels (see “citation list 1B” below). This is consistent with our observation that not all of the substrates that exhibited kinase-dependent changes in their stability were TFs.

Citation List 1A) Examples of DNA-binding proteins/TF’s whose levels are modulated by phosphorylation:

- Phosphorylation-dependent degradation of c-Myc is mediated by the F-box protein Fbw7 (doi:10.1038/sj.emboj.7600217)
- Phosphorylation-dependent ubiquitylation and degradation of androgen receptor by Akt require Mdm2 E3 ligase. (PMID: 12145204)
- Control of lipid metabolism by phosphorylation-dependent degradation of the SREBP family of transcription factors by SCF(Fbw7). (PMID: 16054087)
- Phosphorylation-dependent degradation of p300 by doxorubicin-activated p38 mitogen-activated protein kinase in cardiac cells. (PMID: 15767673)
- HOS, a human homolog of Slimb, forms an SCF complex with Skp1 and Cullin1 and targets the phosphorylation-dependent degradation of IkappaB and beta-catenin. (doi: 10.1038/sj.onc.1202760)
- ATF4 degradation relies on a phosphorylation-dependent interaction with the SCF(betaTrCP) ubiquitin ligase. (PMID: 11238952)
- CDKs promote DNA replication origin licensing in human cells by protecting Cdc6 from APC/C-dependent proteolysis (PMID: 16153703)

Citation List 1B) Examples of non-TF/DNA-binding proteins whose levels are modulated by phosphorylation:

- Phosphorylation-dependent degradation of the cyclin-dependent kinase inhibitor p27 (doi: [10.1093/emboj/16.17.5334](https://doi.org/10.1093/emboj/16.17.5334))
- Ligand-induced internalization and phosphorylation-dependent degradation of growth hormone receptor in human IM-9 cells ([http://dx.doi.org/10.1016/0303-7207\(94\)90187-2](http://dx.doi.org/10.1016/0303-7207(94)90187-2))
- The Phosphorylation-Dependent Regulation of Mitochondrial Proteins in Stress Responses (doi:10.1155/2012/931215)

- 2) In addition to the tendency of TFs and other proteins to undergo phosphorylation-dependent changes in stability, we believe that the final distribution of kinase-dependent changes observed in the first round validation experiments may also have been affected in

another, more subtle way. Specifically, we believe that “mobility shifts” may have been under-represented in the final tally due to the incorporation of a FLAG-tag in the substrates. Indeed, similar to phosphorylation, the presence of the FLAG sequence (DYKDDDDK) introduces a net negative charge of -3 to each substrate. Thus, each FLAG-tagged substrate is likely to be “mobility shifted” relative to its calculated MW at the outset of the experiment, regardless of whether it is phosphorylated or not. Consistent with this notion, our FLAG-tagged substrates tended to migrate at MW’s higher than would be expected based solely on their primary amino acid sequence. As a consequence, phosphorylation events that would normally cause mobility shifts may have been masked, leading these interactions to be designated as “no change”. Moreover, in those instances where kinase-dependent changes in stability caused either the complete disappearance of the substrate (in the case of degradation) or the emergence of a substrate (in the case of stabilization), no conclusions could be drawn about the electrophoretic mobility of the substrate because there is no point of reference.

5. p.15 *"First, we observed that Tyr phosphorylation of PKA, which increased ~2-fold ten minutes after BCR activation, was prevented by pre-treatment of the B cells with the Btk-specific inhibitor, terreic acid (TA) (Figure 5H). This correlated with an increase in the extent of PKA-mediated phosphorylation on a portion of cellular PKA substrates, including ARID3A (Figure 5I,J)". It is not clear to which part of the first sentence "This" refers to.*

RESPONSE: The sentence has been revised to read: "...The observed increase in Tyr phosphorylation on PKA correlated with an increase in the extent of PKA-mediated phosphorylation on a portion of cellular PKA substrates, including ARID3A (Figure 5I,J)" (Page 17).

6. *There is no reference regarding terreic acid being a Btk inhibitor. Is this a specific one? Need support here.*

RESPONSE: Terreic acid is a selective inhibitor of Btk. We have added the following reference: Kawakami et al. (PMID: 10051623) (Page 17).

7. *The normalization process is not sufficiently detailed in the Supplementary Material.*

RESPONSE: We have added more details to the normalization process (Supplementary Methods, Page 3).

8. *Sometimes the authors use the acronym HCLS for Holocarboxylase synthetase, instead of HLCS. A striking example is present on p. 9. The same thing is applicable to Ks-KSI in Supplementary Figure 6-7.*

RESPONSE: “HCLS” has been changed to “HLCS” in the text (Page 10-11).

9. *Supp. Material. No page number.*

RESPONSE: We included page numbers for Supplementary material.

10. *Supp. Material, Saccharomyces cerevisiae is misspelled.*

RESPONSE: We corrected the typo (Supplementary Methods, Page 1).

11. *Supp. Material. Gelman et al. reference is incomplete.*

RESPONSE: Done (Supplementary Methods, Page 3).

12. *It is not clear what the negative set constructed with 10,000 protein pairs is. Is it justifiable to*

use non-kinase proteins in this case? Several types of negative sets should be considered to see what the effects would be.

RESPONSE: In the literature people usually do not report negative cases (e.g. kinase X does not phosphorylate protein Y). Therefore, it is not easy to collect known negative cases as a negative control. In addition, it is not reasonable to include protein pairs without known phosphorylation relationships, because they are often not true negatives. In most cases, they have just not been examined before. To guarantee the protein pairs truly have no KSRs, we include protein pairs without any kinase as a negative control (Supplementary Methods, Page 3).

13. Supp. Material. Statistics about co-expression, colocalization etc. are not reported in the section on Implementation. The authors only mention significant differences or enrichments but numbers need to be provided.

RESPONSE: We provided more statistics on co-expression, co-localization and protein-protein interactions (Supplementary Methods, Page 7).

Reviewer #3 (Remarks to the Author):

The study of Zhang, Zhu and Qian and coworkers describes a very comprehensive screen and follow up of the substrates for 289 human kinases encompassing over 2,656 kinase substrate relationships. The expression, KSRs, analysis pipeline, and follow up with the Btk, PKA pathway relationships represent a very useful new analysis method for defining kinase pathways.

The approach to compare the in vitro KSRs with the published shotgun MS/MS data is a clever idea for large-scale phospho-motif determination. There is an implicit assumption that the phosphorylation detected in vitro holds true in vivo. Can the authors comment on the issue of whether proteins identified as phosphorylated on the arrays by the presence of radioactive labeling matches the sites identified in the shotgun phosphoproteomic databases? This needs clarification because the way it reads now the authors suggest the sites are the same. If the authors could clarify this issue in the text with their current data it would strengthen the manuscript.

RESPONSE: We thank the reviewer for the positive and insightful comments. To answer the question about the phosphorylation sites within the substrates on the microarrays, no, we made no assumptions about the site(s) of phosphorylation on the substrates immobilized on the microarray *a priori*. Instead, to predict consensus phosphorylation motifs using the M3 algorithm, we first asked which substrates are phosphorylated by a given kinase. We then used the phosphoproteomic databases to identify the sites on those substrates that are phosphorylated *in vivo*. Finally, M3 used an iterative approach to identify those sequences that are enriched among the known phosphorylation sites within the *in vitro* substrates. Of course, in some cases, the predicted sites are not necessarily the sites targeted by the kinase-of-interest on a given substrate during the protein microarray experiments. For instance, the true target site(s) might not yet be identified by MS/MS. In this case, the predicted motifs and phospho-acceptor sites can be further refined as the knowledge base grows. That being said, our mutagenesis validation suggested that most of the currently predicted sites are real targets of the upstream kinases. We added some discussion on this issue (Page 19).

It is interesting that in the majority of cases the authors found changes in protein stability in the co-transfection experiments. I am wondering what the reason for this could be. Are the majority of phosphosites in phosphodegrons? Or does overexpression of important kinases like Akt, MAPK, etc. have global effects on the cell that results in changes in the protein expression pattern. Are known oncogenic kinases for example more likely to generate changes in the expression pattern in their system? I think the authors could comment on this issue.

Once these textual issues are addressed I suggest the manuscript be accepted for publication. The authors' have done a remarkable job in assembling a map of phosphorylation sites which will be rapidly utilized by others in the field.

RESPONSE: Regarding the question about the high frequency with which we observe kinase-dependent changes in protein stability, please see our response to reviewer #2, minor point #4. With respect to the second point (i.e., that overexpression of important kinases may have global effects on protein expression), in the context of the experiments conducted in the first stage validation, it seems unlikely that global effects on gene expression would be the root cause of the changes in protein stability that we observe. This is because in our assay each of the substrates is expressed under the control of the same strong, SV40 promoter. Therefore, if a particular kinase affects global gene expression (or even just expression at the SV40 promoter), we would expect to see the same phenotype for each of the substrates for that particular kinase. However, that is not what we observe. Instead, we see that for most kinases, there are some substrates that are stabilized (ST), some that are degraded (D) and some that show no change (NC). For example, overexpression of STK17A led to *i*) a reduction in the levels of FIP1L1, ELL3, and PCBP2, *ii*) an increase in the levels of ASCC1, ENO1 and WHSC2 and *iii*) no change in the levels of ARID3A, IRF5, and MEOX1. Likewise, overexpression of PKA caused *i*) a decrease in the levels of HLCS and NEUROD1, *ii*) an increase in ARID3A, NFATC3, IRF3 and DAXX, and *iii*) no change in CSCD2, SAFB2 and VHL. The same was true for MAPK1 (similar number of D, ST and NC phenotypes) and AKT (4 D, 1 ST and 2 MS).

Reviewer #4 (Remarks to the Author):

In their article "Construction of a High-Resolution Map of Phosphorylation Networks in Humans" Newman et al. present an extensive, experimentally derived kinase-substrate network, conduct an extensive experimental validation, and provide an example of its value for the further elucidation of a well-studied signaling pathway.

With their study, the authors address a pertinent issue for the global analysis of signaling networks. Recent phospho-proteomics methods can measure approximately 20,000 distinct phosphorylation events in cells. However, only for a small minority of these sites the upstream kinases are known, which makes the reconstruction of phosphorylation networks challenging. The presented data set significantly contributes to fill this knowledge gap by providing 3,656 high-quality kinase-substrate relationships (KSRs).

Other studies have addressed this issue, previously. For example, Linding et al. 2007 (NetworKIN) and Miller et al. 2008 (NetPhorest) have predicted kinase-substrate relationships using sequence-based classifiers alone or by also considering the network context. In contrast to these studies, the current study is based on a more extensive experimentally derived data set using protein-array kinase assays.

Highlights of this study

- *The authors present the currently most comprehensive, experimentally-derived data set of kinase-substrate relationships*
- *The authors conducted extensive validation experiments to demonstrate the quality of the derived KSRs*
- *The authors devised a novel algorithm (M3) that identifies position weight matrices from mixed (but enriched) target site populations*
- *The authors use a clever strategy (including their novel M3 algorithm) to identify specific phosphorylation sites from a dataset that (by itself) only resolves general kinase-substrate relationships*

Caveats to keep in mind

- *The tested substrate proteins were selected for a different purpose (as DNA-binding proteins). The potential bias due to this selection should be accounted for, when the data set is used in future analyses*
- *The approach has a relatively high false negative rate of 95% as judged by the recovery of kinase-substrate relationships from a literature set.*

Overall, this study presents a major contribution to our understanding of phosphorylation networks. The KSR data set is extensive and the authors convincingly demonstrate its high quality. Beyond the presented experiments, the KSR data set represents an important resource for future systems-level analyses. With this, I would in principle recommend this manuscript for publication in MSB.

However, first I would ask the authors to address the following issues:

Main text

- Page 6 "Individual protein microarrays, composed of 4,191 unique, full-length human proteins from 12 major protein families (Hu et al., 2009)": The use of protein rather than peptide arrays to measure kinase specificities/activities can be regarded as a major plus of this study - kinase activities are restrained by the substrate accessibility in the folded protein structure and additional protein interactions can play a role. However, this potential benefit is neither analyzed nor discussed. Include a discussion on this topic and analyze the potential benefit of protein arrays for this study. Also include a discussion of potential deficiencies, e.g. effects of improper folding or artificial steric hinderance.

RESPONSE: We thank the reviewer for the insightful comments. As suggested by the reviewer, we discussed the benefit and potential deficiencies of using full-length proteins (Page 19).

- Page 6 "from 12 major protein families (Hu et al., 2009)": The functional group targeted in the original publication was "DNA binding proteins". Thus, the protein list is highly biased toward this protein class. Such a bias could lead to analysis artifacts in subsequent (global) studies using this dataset. To prevent these potential problems, the authors need to more clearly communicate the extent and potential implications of this bias. Extend the discussion on the selection of these proteins, show the distribution of protein classes (and subcellular localization) in comparison with the whole proteome in a main figure, and analyze, whether there is any bias in the recovery of known KSRs comparing the selected protein set with the remaining proteins.

RESPONSE: It is true that the proteins on the microarray are not a homogenous representation of the entire human proteome. The proteins on the microarray are enriched for transcription factors and their co-factors, chromatin-associated proteins, nucleic acid binding proteins, RNA-binding proteins, mitochondrial proteins and kinases. These proteins are known to have phosphorylation-dependent biological activities. Although the proteins families are biased toward these protein classes, it is unlikely that our subsequent global studies are affected by this bias. For example, for the prediction of kinase motifs, as long as there are a sufficient number of substrates for a given kinase, we are able to predict a reliable motif. It is unlikely that the substrates in different protein classes will have different recognition rules for a given kinase. Furthermore, we also predicted the high-resolution map of phosphorylation networks. Since each KSR is considered as independent to other KSRs, our map is unlikely to be altered due to biased protein classes in the substrates. As suggested by the reviewer, we included a supplementary figure to show the distribution of protein classes and subcellular localization. We have added a new figure to the supplemental materials to show the percentage distribution of biological function and cellular localization for each protein family (Page 7, Fig S1).

- Page 7 "Using a P value of 0.05 as a threshold...": the "p value threshold" was estimated based on a permutation of L values of the three selected features. Explain the rational of choosing this p-value estimation approach.

RESPONSE: It is difficult to derive a P value based on analytic approaches, because the distributions of these three features are quite different and discrete. Therefore, we used permutation to estimate the P value. To do this, we permuted the L values of all protein pairs for each of the three features. After this permutation, all the biological correlations inherent in the data have been disrupted, and each protein pair has no relationship with the three L values associated to it. The L value corresponding to the top 5% was selected as cutoff. All the KSRs with L values greater than this cutoff are likely to be biologically relevant (Supplementary Methods, Page 8).

- Page 7 "Second, cross-validation analysis revealed that the Bayesian approach increases the recovery rate of known KSRs over 5-fold": this analysis also suggests a high false negative rate of 95% (only 22 of 421 known KSRs were identified). This should be more clearly stated in the manuscript.

RESPONSE: We included this information in the manuscript. See Discussion section (Page 20).

- Page 8 "the most common change observed was altered protein levels": this observation is really striking and its discussion should be extended, e.g. by discussing the potential mechanistic basis and by comparing the properties of these phosphorylation sites with sites that are known to affect protein stability.

RESPONSE: Please see our responses to Reviewer #2, minor point 4 and Reviewer #3, point 2. We have also included a discussion of this point in the text (see Discussion section, pages 20-21).

- Page 10 "This approach combines our rawKSR data...": Discuss the rationale of selecting the rawKSR rather than the refKSR or comKSR dataset

RESPONSE: The preferred substrate sequence of a kinase is mainly determined by the substrate recognition regions of the kinase and is often characterized in in vitro assays. Therefore, the rawKSRs identified from microarray experiments can be used for motif prediction. We have provided the rationale in the text (Pages 11-12).

- Page 11 "To independently validate the identified phosphorylation motifs, we compared our predicted phosphorylation motifs for 24 kinases to those obtained using a positional scanning peptide library": Extend this comparison to the "Linear Motif Atlas" published by Miller et al. (2008) (<http://netphorest.info>).

RESPONSE: We have added a supplemental figure (Fig. S12) to compare our predicted motifs with those predicted in the "Linear Motif Atlas" published by Miller et al. Because the "Linear Motif Atlas" only provides the motif logos without position weight matrices (PWMs), we cannot make quantitative comparisons between our motifs and theirs. Instead, we evaluated the similarity by visual inspection. The comparison indicated that, for the motifs covered by both datasets, most of them looked very similar. However, our dataset includes more motifs (300) than the Motif Atlas (62). Moreover, because each of our motifs includes a PWM, they can be more easily evaluated in the future.

- Page 12 "what effect mutation of the predicted phospho-acceptor site had on the substrates in the presence of kinase for four selected KSRs.": Explain, how these four KSRs were selected. Were additional KSRs tested?

RESPONSE: These four KSRs were randomly selected for validation. In addition, we also tested the KSRs in the Btk->PKA->ARID3A example.

- Page 12 "A high-resolution map of human phosphorylation networks": Compare these results with the results from the NetworKIN approach (http://networkin.info/version_2_0/search.php). For this, also separately evaluate the 768 known and the 3659 novel KSRs. This will allow the reader to better appreciate the benefits of the presented vs. previously published approaches. Figures

RESPONSE: We compared our network with that derived using the NetworKIN approach. NetworKIN predicted 7,143 site-specific kinase-substrate interactions. After removing those interactions with outdated ENSEMBL IDs for kinases or substrates, 6,338 site-specific kinase-substrate interactions were left. Our high-resolution network predicted 4,417 site-specific kinase-substrate interactions. We compared these two sets with the 1,156 known site-specific kinase-substrate interactions collected from published literature or public databases. Our data recovered 758 known interactions, yielding a true positive rate of 17.2% (758/4417). NetworKIN, however, recovered 48 known interactions, with a true positive rate of 0.82% (48/5864). We included this comparison in the Discussion section (Page 19).

- Fig. 1: Add a short summary legend.

RESPONSE: Done.

- Fig. 2A: The "mobility shift" observed by the authors for the two selected KSRs is not (clearly) visible. This is especially the case for the first pair (HCLS1/NEK10), for which the substrate band is overexposed. For the 2nd pair (PRKACA/EPHA10) a decrease of the lower band is visible, but it is

unclear how reproducible this (minor) effect is. While the (de-) stabilization effects are more convincing, the authors need to provide more evidence for the observed "mobility shifts". The authors might consider Phos-tag acrylamide (<http://www.phos-tag.com/english/>) for more pronounced shifts.

RESPONSE: The HCLS1/NEK10 example has been replaced with another kinase substrate pair (CUEDC2/PRKACA) that also shows a mobility shift. Hopefully this example is clearer. With regard to the 2nd pair (PRKACA/EPHA10) this phenotype was observed on >10 occasions during the first round of validation. Strikingly, it only was observed when the substrate was itself a kinase. We interpreted this as an activation phenotype, a notion that is supported by an in depth analysis of the Btk-PKA pair (e.g. see Fig. 5 E,G,I and Supplemental Fig. 13C).

- Fig. 4B: For some of the tested pairs the effect of substrate mutation is not apparent. For example, the blots for the CBL-S900/MAPK15 pair still show an increase in substrate levels. Overall, it should be made clear, how reproducible these results were. In addition, for the PRKARIA/PRKACA pair substrate mutation affects the protein level by itself. This should be explained.

RESPONSE: With regard to the CBL^{S900A}/MAPK15 results, these experiments have been repeated several times. In each case, a similar ~1.8-fold reduction in the extent of MAPK15-dependent stabilization is observed. In the revised figure, we have included average fold change and standard errors for both WT and MT CBL in the presence of kinase versus the same protein in the absence of MAPK15 (these values are 3.03 +/- 0.03 and 1.88 +/- 0.05, respectively). One possible explanation for the incomplete abolition of MAPK15-dependent stabilization in the CBL^{S900A} mutant may be that a second phosphorylation site in CBL (either for MAPK15 or another kinase activated by MAPK15 overexpression) exists which also contributes to the stabilization of the wild-type protein. With regard to the lower basal levels observed for the PRKARIA^{S77A} mutant protein (in both the presence and absence of exogenous expression of V5-PRKACA), we believe that because endogenous, basally-active PRKACA is unable to phosphorylate the mutant protein, its basal levels are reduced.

Supplement

- Include the high-resolution network as a text or excel file (instead of a pdf file)

RESPONSE: As suggested, we included the high-resolution network as a flat tab-separated text file as supplementary material.

- Include the actual position weight matrices (PWM) derived from the M3 algorithm in addition to their graphical representations

RESPONSE: We also included each PWM in the supplementary material.

- Include figure of all Western Blot results from the first-stage validation studies

RESPONSE: We included the validation results as supplementary material.

- Page 8 "To this end, we collected 1,108 known KSRs from database and literature sources." 1,103 "known KSRs" were used before as the positive training set. Why this difference in the reported numbers?

RESPONSE: We corrected the typo.

- Fig. S4.: explain F/B ratio in legend

RESPONSE: Done.

2nd Editorial Decision

10 February 2013

Thank you again for submitting your work to Molecular Systems Biology. We have now heard back from two referees who accepted to evaluate the study. As you will see, the referees are now supportive and we will be able to accept your paper for publication pending the following points:

1. Please format the reference list to our style (equivalent to The EMBO J).
2. Please include an author contribution section.
3. We would suggest to use the title "Construction of Human Activity-Based Phosphorylation Networks".
4. Please zip compress your cytoscape.cys session file into a Cytoscape.cys.zip file prior uploading it to the tracking system. This way, the .cys extension is re-generated when users download and uncompress the file. It should fix the issue encountered by one of the reviewers.

Referee reports:

Reviewer #2 (Remarks to the Author):

The authors have made all of the necessary changes, which greatly improved the manuscript.

Reviewer #4 (Remarks to the Author):

In their revised version of the manuscript Newman et al. have adequately addressed all my previous comments except for one. Their revised submission does not include a text or Excel file of their "high-resolution map of human phosphorylation networks" (the Cytoscape session of this data did not load in the current Cytoscape version). I would ask the authors to include this data as a supplementary text/Excel file and with this would then recommend their important contribution for publication.

2nd Revision - authors' response

22 February 2013

We are glad to know that our manuscript is accepted for publication in *Molecular Systems Biology*. We believe that we have satisfactorily addressed the points that you raised in the decision letter. Please see our responses below.

We have downloaded the output style of *Molecular Systems Biology* from the Endnote website and used it to reformat the reference list.

We have added an author contribution section before the Acknowledgements section.

Thank you for your title suggestion. We have changed the title accordingly.

We have compressed the "Cytoscape session file" into the Cytoscape.cys.zip file. We hope this will resolve the concern of reviewer # 4.